# Poststroke dendritic arbor regrowth requires the actin nucleator Cobl

Yuanyuan Ji[1], Dennis Koch[1], Jule González Delgado[1], Madlen Günther[2], Otto W. Witte[2], Michael M. Kessels[1]*, Christiane Frahm[2]*, Britta Qualmann[1]*

1 Institute of Biochemistry I, Jena University Hospital–Friedrich Schiller University Jena, Jena, Germany,
2 Hans Berger Department of Neurology, Jena University Hospital, Jena, Germany

* Michael.Kessels@med.uni-jena.de (MMK); Christiane.Frahm@med.uni-jena.de (CF); Britta.Qualmann@med.uni-jena.de (BQ)

## Abstract

Ischemic stroke is a major cause of death and long-term disability. We demonstrate that middle cerebral artery occlusion (MCAO) in mice leads to a strong decline in dendritic arborization of penumbral neurons. These defects were subsequently repaired by an ipsilateral recovery process requiring the actin nucleator Cobl. Ischemic stroke and excitotoxicity, caused by calpain-mediated proteolysis, significantly reduced Cobl levels. In an apparently unique manner among excitotoxicity-affected proteins, this Cobl decline was rapidly restored by increased mRNA expression and Cobl then played a pivotal role in poststroke dendritic arbor repair in peri-infarct areas. In *Cobl* knockout (KO) mice, the dendritic repair window determined to span day 2 to 4 poststroke in wild-type (WT) strikingly passed without any dendritic regrowth. Instead, Cobl KO penumbral neurons of the primary motor cortex continued to show the dendritic impairments caused by stroke. Our results thereby highlight a powerful poststroke recovery process and identified causal molecular mechanisms critical during poststroke repair.

## Introduction

Five million people remain permanently disabled after stroke each year. In the infarct area, stroke leads to a loss of neurons and neuronal network connections due to lack of energy, excitotoxicity, oxidative stress, inflammation, and apoptosis as pathophysiological events [1]. Ischemic stroke caused by middle cerebral artery occlusion (MCAO) accounts for approximately 70% of all infarcts [2,3] and can also be achieved experimentally in rodents [4]. The relative lesion size of survivable human stroke is usually limited to a few percent of the brain [5]. In mice, such damages are very well resembled by 30-minute induced MCAO. By contrast, prolonged paradigms do not resemble survivable human strokes, as they lead to a loss of large parts of the entire hemisphere affected and to significant structural changes at distant or even contralateral sites [5–7].

Ischemic stroke does not only lead to neuronal death and loss of connectivity inside of infarct areas; it also causes a loss of synapses (dendritic spines) adjacent to the infarct area (penumbra) [8–10]. The spine loss is associated with excessive $Ca^{2+}$ influx activating

**Data Availability Statement:** All relevant data are within the paper and its Supporting Information files.

**Funding:** This work was supported by grants from the DFG (Deutsche Forschungsgemeinschaft) to

M.M.K. (KE685/3-2) and to B.Q. (RTG1715) as well as by the IZKF (Interdisziplinäres Zentrum für klinische Forschung des Universitätsklinikums Jena) to B.Q. and C.F. (RTG1715 SP18). The funders had no role in study design, data collection and analysis, decision to publish, or preparation of the manuscript.

**Competing interests:** The authors have declared that no competing interests exist.

**Abbreviations:** CaM, calmodulin; CCA, common carotid artery; CP-1, calpain inhibitor 1; ECA, external carotid artery; Gadph, glyceraldehyde 3-phosphate dehydrogenase; HBSS, Hanks' balanced salt solution; HEK293, Human Embryonic Kidney 293; Hmbs, hydroxymethylbilane synthase; ICA, internal carotid artery; KO, knockout; MCAO, middle cerebral artery occlusion; MIP, maximum intensity projection; NDS, normal donkey serum; qPCR, quantitative PCR; RNAi, RNA interference; ROI, region of interest; RT, room temperature; Tubb3, tubulin beta 3 class III; VEGFD, vascular endothelial growth factor D; WT, wild-type.

numerous downstream effects. Prominent among those is the $Ca^{2+}$-activated calpain-mediated breakdown of synaptic scaffold and receptor proteins, such as spectrin/fodrin, PSD95, and NR2B [11–13]. Importantly, in the penumbra, these synaptic defects are transient and are subsequently repaired, as spines remain somewhat plastic even during adulthood [14]. Dynamics of dendritic spines (also referred to as dendritic remodeling, dendritic plasticity, or synaptic plasticity) is considered as important cellular mechanism for learning but also for compensatory synaptic repair subsequent to stroke [9,10,15–18].

Most dendrites and dendritic branches become stabilized at an age of about 20 days in mice [14], and much less is known about the fate of the dendritic arbor in the penumbra during the acute phase of cerebral ischemia in mice when compared to dendritic spine dynamics. The reason is that the majority of published studies focused on putative long-term effects inside and outside of ischemic areas several weeks or even several months after (often massive) brain damage caused by cerebral ischemia induced by different means in rodents [6,15,19–23]. Other studies tried to identify long-term effects at the contralateral side that may represent compensational processes [24–26] rather than focusing on the ipsilateral penumbra neighboring the infarct area. Recent reports suggested that inside of the penumbra, acute changes in the dendritic arbor occur or can be pharmacologically induced [27,28], respectively.

The morphology of the complex dendritic arbor of neuronal cells is stabilized by microtubules; yet, it is the de novo generation of actin filaments that powers the formation of initial protrusions from dendrites that establishes the dendritic arbor [29]. The Wiskott–Aldrich domain 2-based actin nucleator Cobl [30,31] (gi:32251014) is widely expressed in the brain [32] and was demonstrated to be critical for dendritic branching of developing hippocampal neurons [30] and of Purkinje cells [32] together with accessory machinery [32–34]. Interestingly, the functions of the actin nucleator Cobl hereby show regulations by arginine methylation by PRMT2 [35] and by multiple $Ca^{2+}$/calmodulin (CaM)-mediated mechanisms directly converging onto Cobl [36].

Here, we show that induction of ischemic stroke in mice leads to a rapid degradation and a subsequent reexpression of the actin nucleator Cobl. Cobl degradation is excitotoxicity-mediated. By analyzing *Cobl* knockout (KO) mice [37], we furthermore demonstrate that Cobl is crucially involved in a process of regrowth of the dendritic arbor, which we unveil to occur in the penumbra in a narrow time window from day 2 to day 4 after ischemic stroke. With Cobl-dependent poststroke dendritic arbor regrowth, our work adds a powerful cell biological process inside of peri-infarct areas that represents an acute and long-range mechanism of poststroke repair.

## Results

### Degradation of the actin nucleator Cobl upon MCAO

The actin nucleator Cobl [30,31] is regulated by $Ca^{2+}$/CaM at physiological $Ca^{2+}$ levels [36]. Excitotoxicity situations, such as massive neurotransmitter releases upon stroke, lead to pathophysiologically high $Ca^{2+}$ levels. Strikingly, quantitative biochemical analyses of brain lysates of mice subjected to ischemic stroke by MCAO (**Fig 1A–1P**) showed a reduction of Cobl protein levels at the ipsilateral side. In comparison to intrinsic contralateral controls, ipsilateral Cobl protein levels declined by about 25% 3 hours and 6 hours after MCAO (**Fig 1A, 1B, 1E and 1F**). Unchanged levels of β3-tubulin demonstrated that the observed Cobl loss did not merely reflect neuronal degradation in the stroke-affected area of the striatum per se. Instead, the effect seemed to represent a widespread targeted degradation of the actin nucleator Cobl (**Fig 1A–1D and 1I–1L**). In line, the levels of an excitotoxicity-induced 140-kDa spectrin/fodrin fragment [38,39] increased upon MCAO. Similar to the Cobl decline, the MCAO-

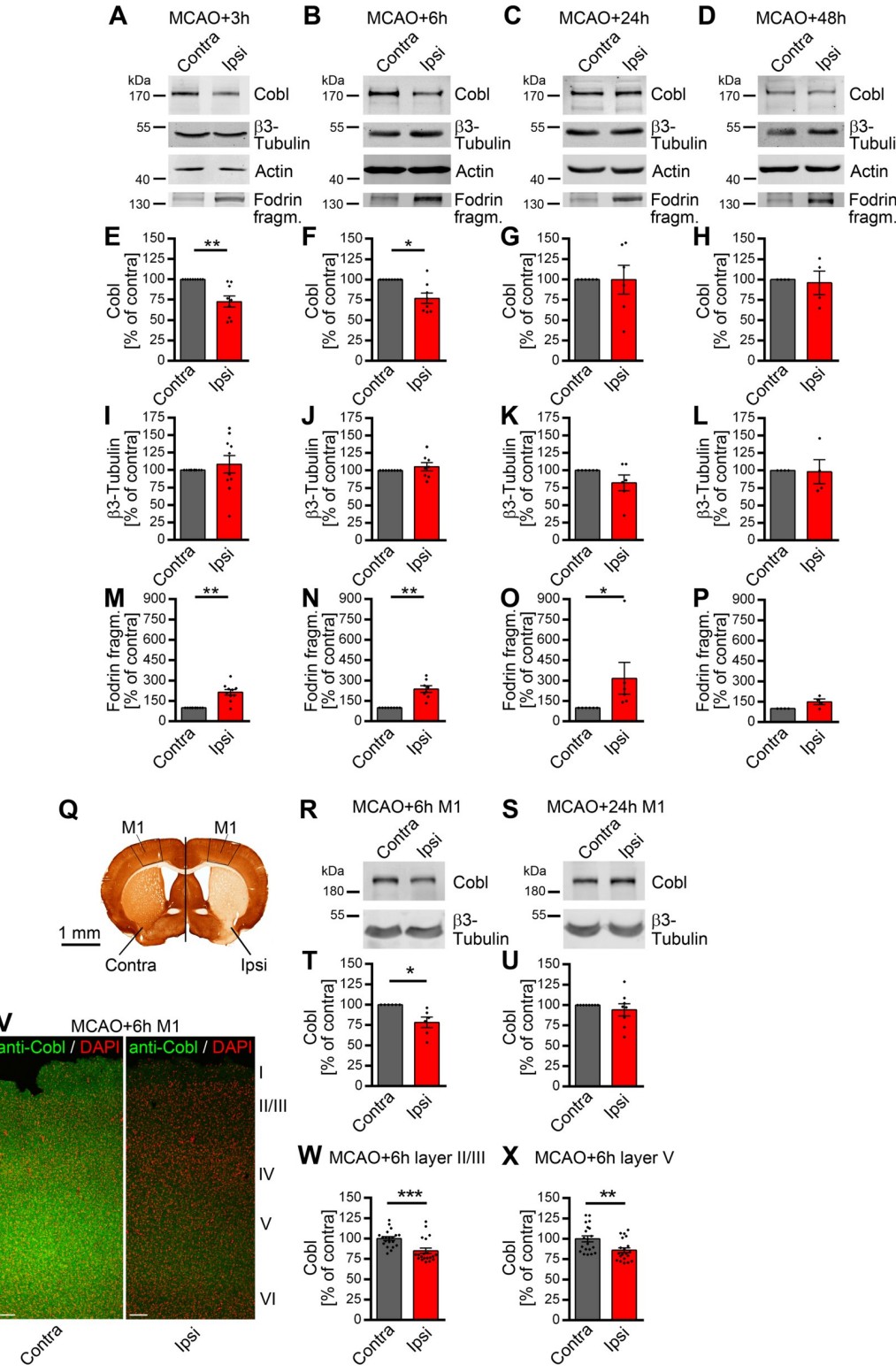

**Fig 1. The actin nucleator Cobl decreases significantly in the ipsilateral hemisphere and in the primary motorcortex (M1) during the first hours after ischemic stroke induced by MCAO. A–P**, Representative western blot images (**A–D**) and quantitative analyses (**E–P**) of Cobl (**A–H**), β3-tubulin (**A–D, I–L**) and a proteolytic spectrin/fodrin fragment of 140 kDa (**A–D, M–P**) in the ipsilateral part of a middle (+0.8 and −1.2 mm to bregma) brain segment in percent of immunosignals at the corresponding contralateral side after different times of reperfusion after 30-minute MCAO. Note

the significant decline of Cobl protein levels at 3 hours and 6 hours after MCAO in the ipsilateral side (**A, B, E, F**). Anti-actin and anti-β3-tubulin immunoblotting signals served as loading controls and for normalization, respectively. **Q**, Anti-MAP2 immunostained coronal brain section with the primary motorcortex (M1) marked (black frame) at both the ipsilateral and contralateral side (dividing line) of brains of WT mice subjected to MCAO. Bar, 1 mm. **R–U**, Representative anti-Cobl and anti-β3-tubulin immunoblotting images (**R, S**) and quantitative analyses (**T,U**) of Cobl levels normalized to those of β3-tubulin and expressed as percent of respective contralateral values. **V**, Representative immunofluorescence images of contralateral M1 (left panel) and ipsilateral M1 (right panel) at 6 hours reperfusion time after 30-minute MCAO stained with anti-Cobl antibodies (green) and with DAPI (red (false color to improve visibility)). Bars, 100 μm. **W–X**, Quantitative analyses of anti-Cobl immunosignal intensities measured in the contralateral and ipsilateral side normalized to the average contralateral intensity in layer II/III (**W**) and in layer V (**X**), respectively. For Cobl levels in brain hemisphere segments (+0.8 and −1.2 mm to bregma), $n_{MCAO+3h}$ = 9; $n_{MCAO+6h}$ = 8; $n_{MCAO+24h}$ = 6; $n_{MCAO+48h}$ = 4 biologically independent brain samples. For fodrin fragment and β3-tubulin levels, $n_{MCAO+3h}$ = 10; $n_{MCAO+6h}$ = 8; $n_{MCAO+24h}$ = 6; $n_{MCAO+48h}$ = 4 biologically independent brain samples. For Cobl levels in M1, $n_{MCAO+6h}$ = 6; $n_{MCAO+24h}$ = 8 biologically independent M1 tissue samples. For Cobl immunofluorescence in layers II/III and V in M1, $n$ = 20 ROIs each (from 4 sections and 2 animals). Data represent mean ± SEM presented as bar plots overlaid with all individual data points. Statistical significance calculations, Wilcoxon signed rank test (**E–P, T, U**), Mann–Whitney test (**W, X**). $^*P < 0.05$; $^{**}P < 0.01$; $^{***}P < 0.001$. The numerical data underlying this figure can be found in S1 Data. MCAO, middle cerebral artery occlusion; ROI, region of interest; WT, wild-type.

induced spectrin/fodrin degradation also showed a fast onset (**Fig 1A–1D and 1M–1P**). The ischemia-induced 140-kDa spectrin/fodrin fragment was still found to be significantly elevated even at 24 hours after MCAO (**Fig 1O**).

Remarkably, the reduced Cobl levels were rapidly restored to levels resembling contralateral values. 24 hours and 48 hours after MCAO, no differences in Cobl levels were observed anymore (**Fig 1C, 1D, 1G and 1H**).

## MCAO leads to Cobl degradation followed by an efficient Cobl level recovery in the motor cortex M1

In order to directly address whether Cobl levels indeed responded to ischemic stroke conditions in wider areas of the brain, i.e., beyond the infarct area itself, and may therefore be of thus far unrecognized importance for the pathophysiology of stroke and/or subsequent repair, we conducted further MCAO experiments and isolated specifically the ipsilateral primary motor cortex (M1)—a physiologically important part of the cortex that is adjacent to but not inside the infarct area and thereby belongs to the penumbra (**Fig 1Q**). Quantitative immunoblotting analyses of dissected M1 tissues clearly revealed the decline of Cobl levels at the ipsilateral side to about 75% of the corresponding contralateral control values of the respective animals 6 hours after MCAO (**Fig 1R and 1T**). Thus, the decline of Cobl in the whole brain hemisphere (**Fig 1A–1H**) can also clearly be observed in the penumbral M1 (**Fig 1Q–1U**).

Also, the recovery of Cobl expression levels at 24 hours after MCAO was observed in quantitative immunoblotting analyses of M1. Twenty-four hours after MCAO, Cobl levels were restored to about 100% of the control value (**Fig 1S and 1U**).

Immunofluorescence analyses of brain sections also demonstrated a clear MCAO-induced ipsilateral decline of Cobl levels in the M1 6 hours post-MCAO (**Fig 1V**). Even without background subtractions applied, Cobl level declined in both layer II/III and in layer V in almost the same order of magnitude as determined by biochemical analyses in M1 and in the central hemisphere segment, respectively. The decline of Cobl was highly statistically significant and about equally strong in layer II/III and in layer V (**Fig 1W and 1X**).

Thus, Cobl protein levels in the cortex show surprisingly high short-term dynamics; these dynamics are related to stroke and can explicitly also be detected in the pyramidal cell layers II/III and V of M1.

## Cobl degradation upon stroke is mirrored by glutamate-induced excitotoxicity brought about by NMDA receptors, high $Ca^{2+}$ levels, and calpain activity

The decreased blood flow to parts of the brain in ischemic stroke leads to a lack of oxygen and energy and therefore to a massive overrelease of glutamate and to dramatic increases of intracellular $Ca^{2+}$ levels [1]. In line with the observed degradation of Cobl in cortex samples of mice subjected to MCAO, cultures of cortical neurons subjected to increasing durations of stimulation with levels of glutamate causing excitotoxicity [40] also showed a clear and very rapid decline of Cobl protein levels in relation to unchanged actin and β3-tubulin levels (**Fig 2A and 2B**).

NR2B and PSD95 are targeted by excitotoxicity processes [11,13]. The Cobl decline was more rapid and more severe than the declines observed for the established excitotoxicity-induced degradation targets NR2B and PSD95 (**Figs 2A–2C and S1A**).

In contrast to the actin nucleator Cobl and to the postsynaptic components NR2B and PSD95, quantitative anti-Arp3 evaluations did not show any declines in Arp2/3 complex levels (**Figs 2A and S1B**). This was somewhat surprising as MCAO is known to modulate dendritic spines [9,10], and the Arp2/3 complex and modulators of its activity, respectively, have been thoroughly established as important for postsynaptic F-actin organization shaping dendritic spines in multiple ways [41–45]. On the other hand, only very few proteins, such as NR2B and PSD95 [11,13], have thus far been identified as directly responsive to excitotoxicity. Arp3 may be protected from proteolytic degradation by its compact fold and its tight interactions with the other Arp2/3 complex components as well as with actin.

Cobl protein levels declined upon excitotoxicity paradigms in both cultures and animals. Excitotoxicity processes are linked to high intracellular $Ca^{2+}$ levels. Consistently, the application of different protease inhibitors showed that the Cobl decline was brought about by the $Ca^{2+}$-activatable protease calpain (**Figs 2D and 2E and S1C**). Both calpain inhibitors, calpain inhibitor 1 (CP-1), and Calpeptin, did not only suppress the occurrence of spectrin/fodrin fragments upon glutamate application but also effectively inhibited the Cobl decline (**Fig 2D and 2E**). Chloroquine and Lactacystin, by contrast, did not suppress the excitotoxicity-induced decline of Cobl protein levels (**Fig 2D and 2E**). Thus, the decline of Cobl did neither reflect a lysosomal nor a proteasomal degradation but was calpain mediated.

Reconstitutions with immunoprecipitated Cobl N and C-terminal halves, respectively, and rising calpain concentrations clearly demonstrated that calpain can indeed lead to a rapid Cobl destruction (**Fig 2F**). A further mapping of calpain cleavage sites revealed that even the use of smaller parts of Cobl (GFP-Cobl$^{1-408}$, GFP-Cobl$^{406-866}$, GFP-Cobl$^{750-1005}$, and GFP-Cobl$^{1001-1337}$) still showed effective degradations into multiple, GFP-containing fragments (**S1D Fig**). The multitude of calpain cleavage sites suggest a complete loss of Cobl functions when Cobl is digested by calpain.

Also in brain extracts, Cobl was susceptible to $Ca^{2+}$-triggered degradation. Importantly, this degradation of endogenous Cobl again was suppressible by calpain inhibitor (**Fig 2G**).

We next addressed the neuronal signal transduction pathways leading to the calpain-mediated Cobl digestion. CNQX, a powerful inhibitor of AMPA and Kainate types of glutamate receptors, did not suppress the excitotoxicity-mediated Cobl decline. By contrast, AP-5, MK801, and also Ifenprodil, which generally interfere with NMDARs, inhibit open NMDARs and explicitly block the NR2B subunit of NMDARs, respectively, all completely suppressed Cobl degradation caused by prolonged incubation with 40 μM glutamate (**Figs 2H and 2I and S1E** for β3-tubulin). Thus, the glutamate-induced Cobl degradation is mediated by the activation and opening of NR2B-containing NMDARs.

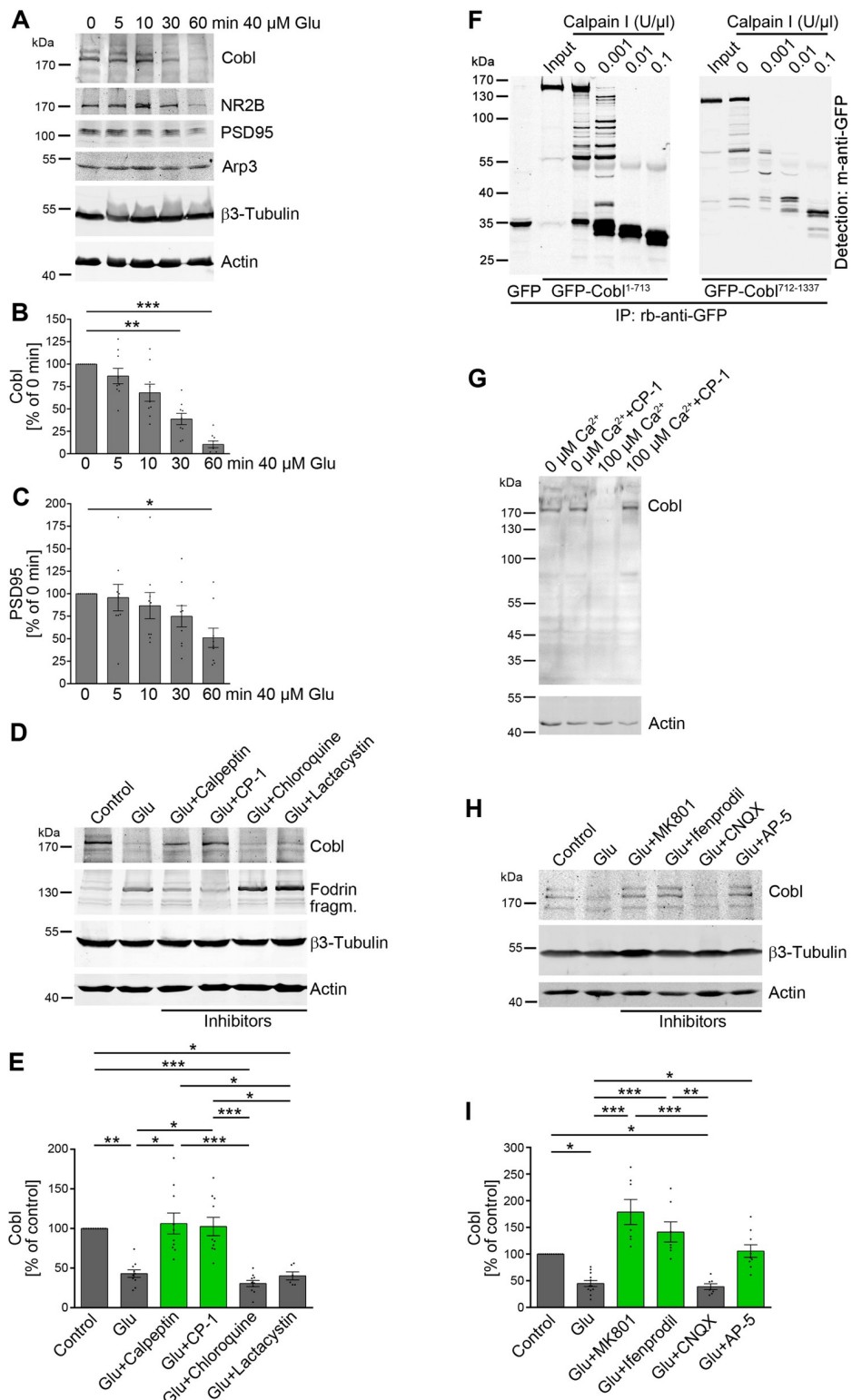

**Fig 2. Cobl degradation upon stroke is mirrored by glutamate-induced excitotoxicity brought about by NMDA receptors, high Ca²⁺ levels and calpain activity.** (**A–C**), Quantitative immunoblotting analyses of cultures of cortical neurons (DIV15) subjected to different durations of incubation with 40 μM glutamate (Glu). **A**, Representative blots. **B, C**, Excitotoxicity-induced decline of anti-Cobl (**B**) and anti-PSD95 (**C**) immunosignals. *n* = 9 independent assays. **D, E** Proteolytic pathways underlying the glutamate-induced (30 minutes) Cobl decline, as shown by application of

inhibitors against calpain (Calpeptin, CP-1), lysosomal degradation (Chloroquine) and proteasomal proteolysis (Lactacystin), respectively, in quantitative immunoblotting analyses. $n_{Control} = 10$, $n_{Glu} = 10$, $n_{Glu+Calpeptin} = 10$, $n_{Glu+CP-1} = 10$, $n_{Glu+Chloroquine} = 10$, $n_{Glu+Lactacystin} = 6$ biologically independent samples. **F**, Anti-GFP immunoblotting analyses of GFP-Cobl$^{1-713}$ and GFP-Cobl$^{712-1337}$ expressed in HEK293 cells, immunoisolated with anti-GFP antibodies (input) and incubated without (0) and with calpain I (10 minutes, 25°C). GFP is shown for size comparison. Note that 0.001 U/µl was sufficient for efficient Cobl digestion by calpain. **G**, Immunoblotting of brain extracts incubated (15 minutes, 4°C) with 0 and 100 µM $Ca^{2+}$ and with and without the calpain inhibitor CP-1. Anti-actin signals serve as loading controls. **H, I**, NMDAR activity is required for the Cobl decline caused by excitotoxicity (40 µM glutamate, 30 minutes) in DIV15 cortical neurons, as shown by inhibitors against AMPA and kainate receptors (Glu+CNQX), NMDARs (Glu+AP-5), open NMDARs (Glu+MK801), and NR2B subunits of NMDA receptors (Glu+Ifenprodil), respectively, in immunoblotting (**H**) and quantitative analyses (**I**). Anti-actin and anti-β3-tubulin immunoblotting signals served as loading controls and for normalization, respectively (**A, D, H**). $n_{control} = 12$, $n_{Glu} = 12$, $n_{Glu+MK801} = 7$, $n_{Glu+Ifenprodil} = 7$, $n_{Glu+CNQX} = 7$, $n_{Glu+AP-5} = 9$ biologically independent samples. Data represent mean ± SEM presented as bar plots overlaid with all individual data points. Statistical significance calculations, 1-way ANOVA with Dunn posttest (**B,C,E,I**). $^*P < 0.05$; $^{**}P < 0.01$; $^{***}P < 0.001$. The numerical data underlying this figure can be found in **S2 Data**. CP-1, calpain inhibitor 1.

Serial section analyses of brains of *Cobl* KO mice [37] subjected to 30-minute MCAO showed that neither the ischemia-induced acute loss of Cobl we detected in the central brain segment and in the M1 nor the subsequent rapid restoration of normal Cobl levels in these brain parts had any influence on the survival of neurons within the infarct area in the striatum (**S1F–S1H Fig**).

With about 5% of the whole brain affected, *Cobl* KO mice had infarct volumes comparable to those of wild-type (WT) mice (**S1F–S1H Fig**). This extend of lesional damage caused by 30-minute MCAO in both genotypes also reliably mirrored the range of survivable human stroke, which usually also represents about 5% of the total brain [5].

## A transient increase of *Cobl* mRNA precedes the rapid recovery of Cobl protein levels after its poststroke decline

The observed rapid recovery of ipsilateral Cobl protein levels 24 hours after MCAO (**Fig 1**) suggested active counteractions of the stroke-affected brain to rapidly replenish the actin nucleator Cobl for some important, yet unknown function(s). This compensation mechanism should be reflected by increased mRNA levels at specifically the ipsilateral, i.e., the stroke-affected side, prior to full restoration of normal Cobl protein levels at 24 hours after MCAO. We indeed observed a statistically highly significant increase of *Cobl* mRNA levels at the ipsilateral side at 2 different time points prior to restoration of Cobl protein levels to contralateral control levels at 24 hours after MCAO (**Fig 3A**).

The increase of *Cobl* mRNA levels in the ipsilateral hemisphere of the brain was detected as early as 6 hours after MCAO (**Fig 3A**), i.e., during a time when Cobl protein levels still were strongly negatively affected (**Fig 1B and 1F**). The increase of *Cobl* mRNA 6 hours after MCAO was validated using 2 additional primer pairs. The results were fully consistent irrespective of primers and normalizations used (**Figs 3A and S2A and S2B**). The increase of *Cobl* mRNA levels after MCAO remained equally strong at 12 hours (**Fig 3A**). It then ceased 24 hours after MCAO (**Fig 3A**) when also Cobl protein levels reached a level similar to the contralateral control side (**Fig 1C and 1G**).

The increase of *Cobl* mRNA levels reflected changes in especially neurons, as it occurred upon normalization to *Gapdh* and *Hmbs* mRNA levels (ubiquitously expressed housekeeping genes) but also upon normalization to *Tubb3* mRNA levels (**Figs 3A and S2A and S2B**).

Interestingly, although NR2B, PSD95, and spectrin 2/fodrin also are targets of proteolytic processes triggered by excitotoxicity [11–13], the mRNA of none of these components showed any transient increase of expression levels after MCAO (**Fig 3B–3D**). The levels of *CaM 1* and

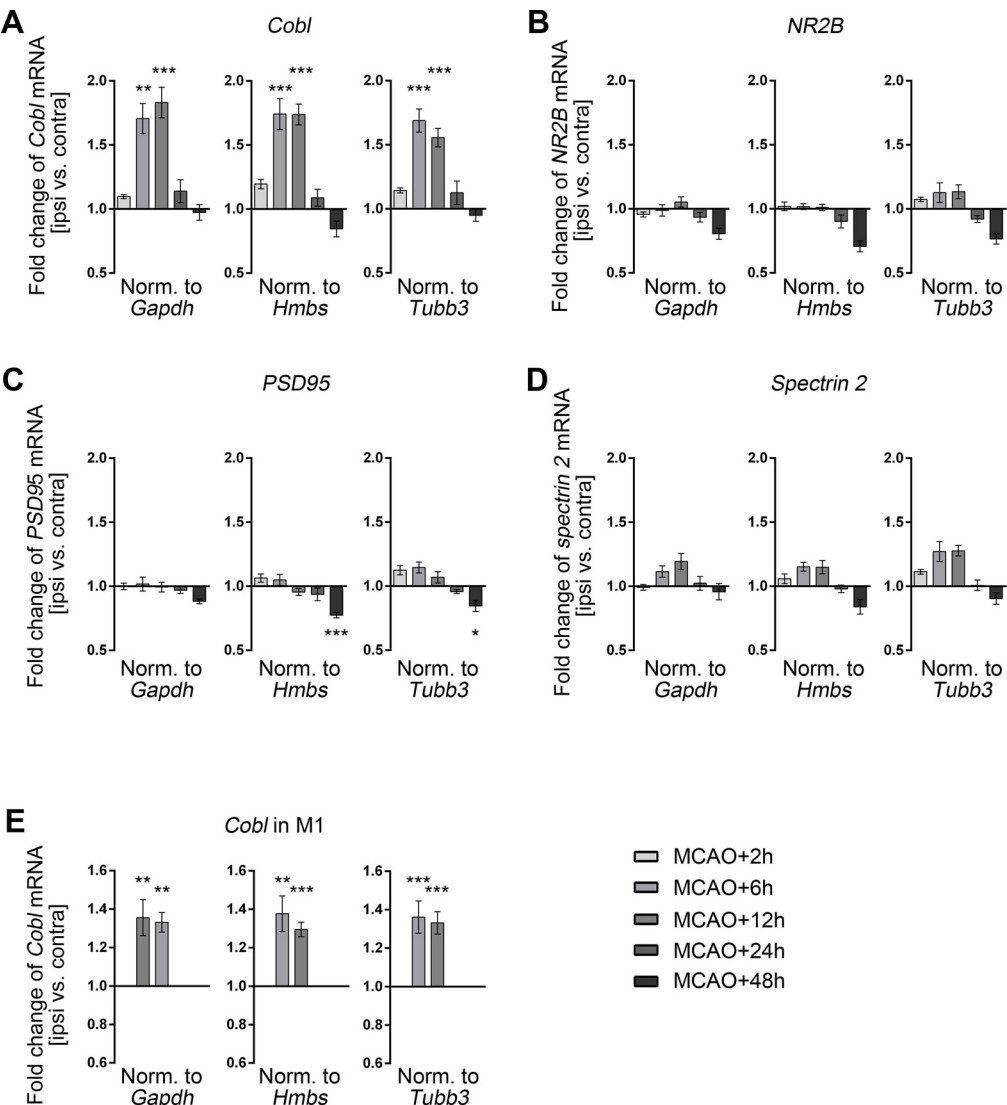

**Fig 3. qPCR analyses demonstrate a transient increase of *Cobl* mRNA levels following the decline of Cobl protein levels upon MCAO. A–D**, Fold change of mRNA levels of *Cobl* (**A**), *NR2B* (**B**), the scaffolding proteins *PSD95* (**C**), and *spectrin 2* (**D**) determined in the middle segment of the brain (+0.8 and −1.2 mm to bregma) by qPCR after 2 hours, 6 hours, 12 hours, 24 hours, and 48 hours of reperfusion after 30-minute MCAO. Data represent the differences of *Cobl*, *NR2B*, *PSD95*, and *spectrin 2* mRNA levels (ipsi versus contra) normalized to *Gapdh* (left panel), to *Hmbs* (middle panel), and to the neuronally expressed gene *Tubb3* (right panel), respectively. Brain samples from 6 to 9 mice were analyzed for each time point. **E**, Fold changes of *Cobl* mRNA levels in M1 tissue samples (ipsi versus contra) at the times of up-regulation identified above (6 hours and 12 hours after MCAO). The data were again normalized against 3 different genes (*Gapdh*, *Hmbs*, and *Tubb3*). **A**, $n_{MCAO+2h} = 8$; $n_{MCAO+6h} = 8$; $n_{MCAO+12h} = 9$; $n_{MCAO+24h} = 6$; $n_{MCAO+48h} = 6$ mice. **B–D**, $n_{MCAO+2h} = 7$; $n_{MCAO+6h} = 8$; $n_{MCAO+12h} = 7$; $n_{MCAO+24h} = 6$; $n_{MCAO+48h} = 6$ mice. **E**, $n_{MCAO+6h} = 7$; $n_{MCAO+12h} = 8$. Data represent mean ± SEM. Statistical significances (ipsi versus contra) were calculated using 1-way ANOVA with Sidak posttest. $^*P < 0.05$; $^{**}P < 0.01$; $^{***}P < 0.001$. The numerical data underlying this figure can be found in **S3 Data**. MCAO, middle cerebral artery occlusion; qPCR, quantitative PCR.

*calpain 1* mRNA—i.e., of 2 factors that dictate the activity and the functionality, respectively, of the actin nucleator Cobl—also did not change at any examined time point subsequent to MCAO (**S2C and S2D Fig**). The transient increase of *Cobl* mRNA levels subsequent to the exitotoxicity-mediated Cobl digestion by calpain and in advance to the rebound of Cobl protein levels thus seems to be a unique compensatory response to ischemic stroke.

This raised the question whether this stroke response could indeed be observed in penumbral areas, such as the M1 similar to the changes in Cobl protein levels (**Fig 1**). We therefore expanded our Cobl analyses in M1 tissue by quantitative PCR (qPCR) experiments at both time points of elevated *Cobl* mRNA levels using 2 independent primer pairs (**Figs 3E and S2E**). *Cobl* mRNA showed clear up-regulations in both 6 hours and 12 hours post-MCAO tissue samples of M1 (**Figs 3E and S2E**).

Thus, both the $Ca^{2+}$-, calpain- and NMDA receptor–mediated decline of Cobl as well as its fast recovery to normal levels driven by a transient increase in *Cobl* mRNA were phenomena observable in neurons of a penumbral cortex area, the M1.

## MCAO leads to strong reductions of dendritic arbor complexity in layers II/III and V of the primary motor cortex (M1)

Besides axons with their presynapses and dendritic spines harboring postsynapses, the dendritic arbor itself is also a major structural element in neuronal wiring in the brain. As the formation of new synaptic contacts subsequent to stroke-dependent loss [9,10] may only be one aspect in the penumbra that compensates for the functions of entire brain regions lost upon ischemic stroke and recently also acute penumbral dynamics of the dendritic arbor were reported after ischemic stroke [27,28], we analyzed the dendritic arbor of the ipsilateral M1.

In order to obtain reliable data addressing infarct-related dendritic changes, we established a procedure that enabled us to evaluate the core infarct area caused by 30-minute MCAO in the striatum of each individual mouse by anti-MAP2 immunostaining and to in parallel analyze morphologies of individual neurons in a defined area adjacent to the damage zone, the penumbra (represented by the primary motor cortex (M1)) using neighbored coronal brain sections (**S3A and S3B Fig**).

Detailed morphometric analyses of both layer II/III (**Fig 4A–4C**) and layer V neurons of the ipsilesional M1 (**Fig 4D–4F**) unveiled that MCAO leads to dramatic dendritic arborization defects when compared to sham-treated animals, which also underwent anesthesia and the surgical procedures but without MCAO (**Fig 4A–4N**). Both the number of dendritic branching points and the number of terminal points per neuron declined subsequent to MCAO at specifically the ipsilateral side when brains were analyzed after 24 hours reperfusion (**Fig 4G, 4H, 4K and 4L**).

In cortex layer II/III, neurons even showed these dramatic impairments already after 6 hours (**Fig 4G and 4H**). The MCAO-mediated defects in layer V neurons developed slower and were observable at 24 hours (**Fig 4K and 4L**). Similar defects and onsets were observed when the entire length of the dendritic arbor was examined in layer II/III and layer V of the cortex (**Fig 4I and 4M**). Also, Sholl analyses of the dendritic complexity showed corresponding reductions of the dendritic arbor (**Fig 4J and 4N**).

These ischemic stroke–induced defects in dendritic organization of neurons in both analyzed areas of the motor cortex were restricted to the ipsilesional side. For all 4 parameters determined, the data obtained from the contralateral side of the same MCAO animals were indistinguishable from those of sham-treated mice (**S3C–S3M Fig**). These data are in line with in vivo imaging data that also did not detect any obvious remapping at the contralateral side after stroke [46].

## As early as 4 days after MCAO, the discovered dendritic arborization defects caused by MCAO were mostly compensated

We next evaluated whether the massive MCAO-induced impairment in dendritic arborization we observed in the M1 (**Fig 4**) was permanent and part of the lesion-based disabilities caused

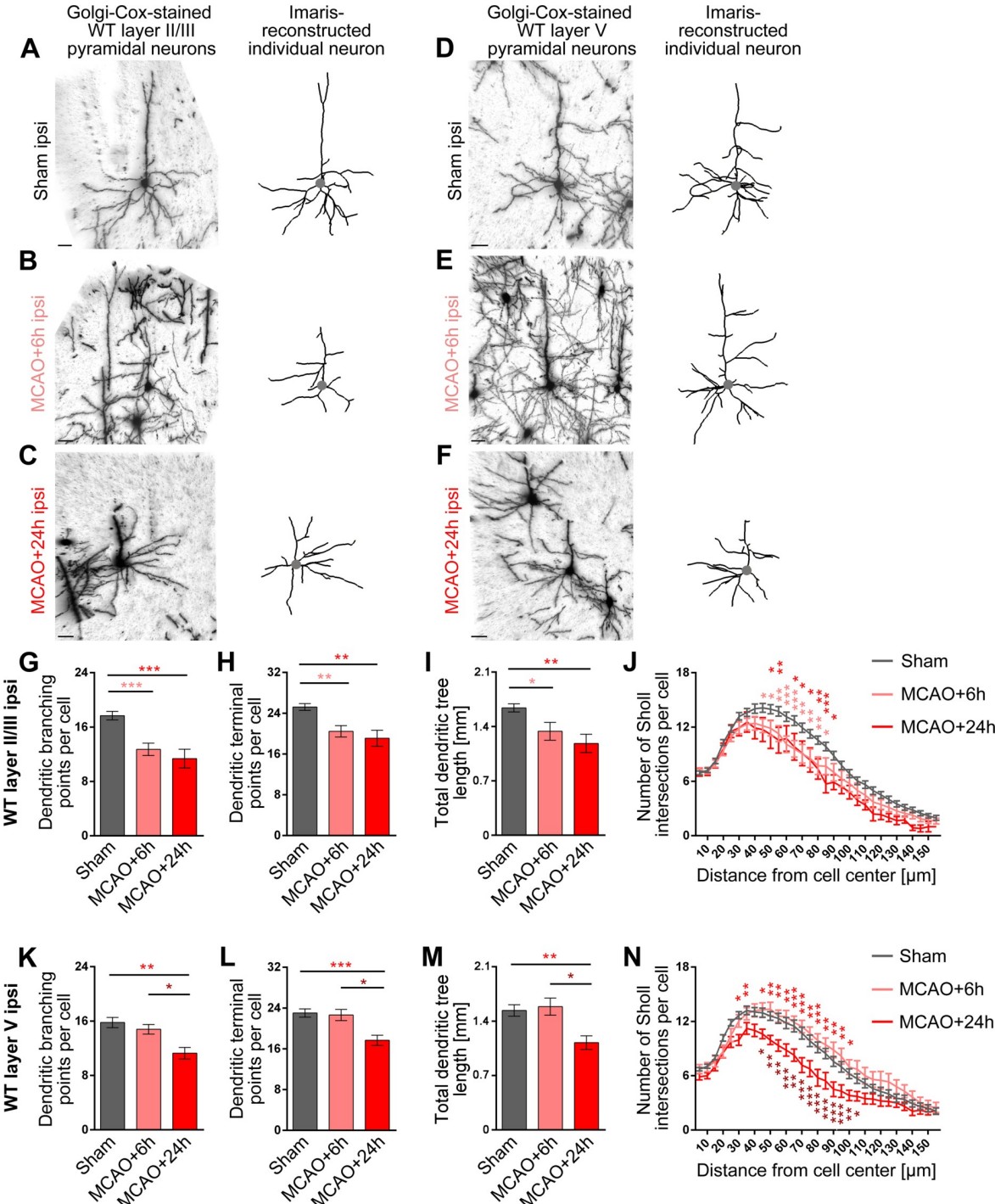

**Fig 4. Loss of dendritic arbor complexity subsequent to ischemic stroke in ipsilateral layer II/III and layer V neurons of the motor cortex. A–F**, Representative images of Golgi–Cox-stained (left panels) and individual Imaris-reconstructed (right panels) layer II/III (**A–C**) and layer V (**D–F**) pyramidal neurons of M1 from the ipsilateral side of sham-treated control mice (**A, D**) and of mice subjected to 30-minute MCAO analyzed after 6 hours (**B, E**) and 24 hours (**C, F**) reperfusion times. The position of the cell bodies are marked by a gray dot. Scale bars, 30 μm. **G–N**, Quantitative determinations of dendritic branching points, terminal points, total dendritic tree length, and Sholl intersections of dendritic trees in layer II/III (**G–J**) and layer V (**K–N**). Note that all parameters of dendritic complexity at the ipsilateral side decreased subsequent to MCAO and that layer II/III neurons responded faster, as they already show the decreased dendritic arborization at 6 hours after MCAO (for contralateral data, see **S3 Fig**). Layer II/III: $n_{Sham}$ = 60; $n_{MCAO+6h}$ = 22; $n_{MCAO+24h}$ = 11 neurons. Layer V: $n_{Sham}$ = 47; $n_{MCAO+6h}$ = 19; $n_{MCAO+24h}$ = 18 neurons from 3 mice for each MCAO group and 6 mice for the sham control (ipsi). Quantitative data represent mean ± SEM. Statistical significance calculations, 1-way ANOVA with Tukey posttest (**G–I, K–M**) and 2-way

ANOVA with Sidak posttest for Sholl analysis (**J, N**), respectively. $^*P < 0.05$; $^{**}P < 0.01$; $^{***}P < 0.001$. The numerical data underlying this figure can be found in **S4 Data**. MCAO, middle cerebral artery occlusion; WT, wild-type.

by stroke or whether it would at some point become repaired by some thus far unidentified form of dendritic dynamics, which may be inducible by ischemic stroke in mature neurons. We therefore analyzed the morphologies of pyramidal neurons in the M1 of WT mice at 4 days and 7 days after MCAO in direct comparison to the defects observed at 24 hours and to sham controls in a blinded manner (**Fig 5A–5D**). Strikingly, in neurons of the ipsilateral layer II/III, all defects observed at 24 hours after MCAO were fully compensated for at day 4 and 7 (**Fig 5E–5J**). The dendritic branching points at day 4 and 7 after MCAO exactly were at the levels of sham animals (**Fig 5E**). Likewise, dendritic terminal points were back at control levels (**Fig 5F**). Total dendritic tree length and Sholl analyses also showed a full restoration of normal dendritic arborization at 4 days and 7 days after MCAO (**Fig 5G–5J**).

Both apical and basal parts of the dendritic arbor of layer II/III neurons showed about equally strong defects (**S4A–S4K Fig**). Dendritic branch points and total dendritic length were reduced by about 30% (**S4D, S4F, S4I and S4K Fig**). In both apical and basal dendrites, the terminal point numbers declined by 25% to 30% but failed to reach statistical significance for the apical dendrites (**S4E and S4J Fig**). Also, the restoration of dendritic arbor complexity was as effective in apical dendrites as it was in basal dendrites (**S4D–S4K Fig**).

Layer V neurons, which showed a slower onset of dendritic defects (**Fig 4**), showed a similarly clear repair of MCAO-induced impairments (**Fig 5K–5P**). Only the number of dendritic branching points still showed some reduction at day 4 (**Fig 5K**). At day 7 after MCAO, all dendritic morphology parameters quantitatively analyzed in MCAO-treated animals were no longer significantly different from those obtained from sham animals (**Fig 5K–5M and 5P**).

Together, our evaluations at the different time points unveiled that the dendritic arborization defects, which we detected upon MCAO in both layer II/III and layer V of the M1, were fully compensated for by a dendritic regrowth process, which gave rise to apparently normally branched and sized dendritic trees and which occurred in both layer II/III and layer V.

Examinations at the contralateral side showed that the ischemic stroke–triggered repair process in M1 is restricted to the defective ipsilateral side and is not a brain-wide phenomenon of dendritic growth induction. At the contralateral side, all parameters remained at sham levels during all time points analyzed (**S5 Fig**).

Severe ischemic stroke models, such as 60-minute MCAO or even a permanent blockage of blood flow in mice, being rather unrelated to survivable human strokes, would not be suitable to study repair processes in the penumbra, as surviving animals usually lose large parts of the entire hemisphere affected and show significant structural changes at distant sites or even at the contralateral side of the brain [5,6], and the penumbra is small or not existing [7]. We did not observe any structural alterations at the contralateral side (**S5 Fig**) but specifically found cellular alterations in the penumbra at the ipsilateral side (**Figs 4** and **5**). This demonstrated that our animal stroke model (30-minute MCAO) reliably mirrored the range of survivable human stroke and thus was suitable to unveil repair processes in peri-infarct areas.

## The actin nucleator Cobl promotes and is critical for dendritic arborization in cortical neurons

The defects caused by MCAO in pyramidal neurons in cortical layers II/III and V of the murine M1 seemed somewhat related to those observed for Cobl loss of function in immature rat hippocampal neurons in culture [30,34]. We therefore addressed whether Cobl may also

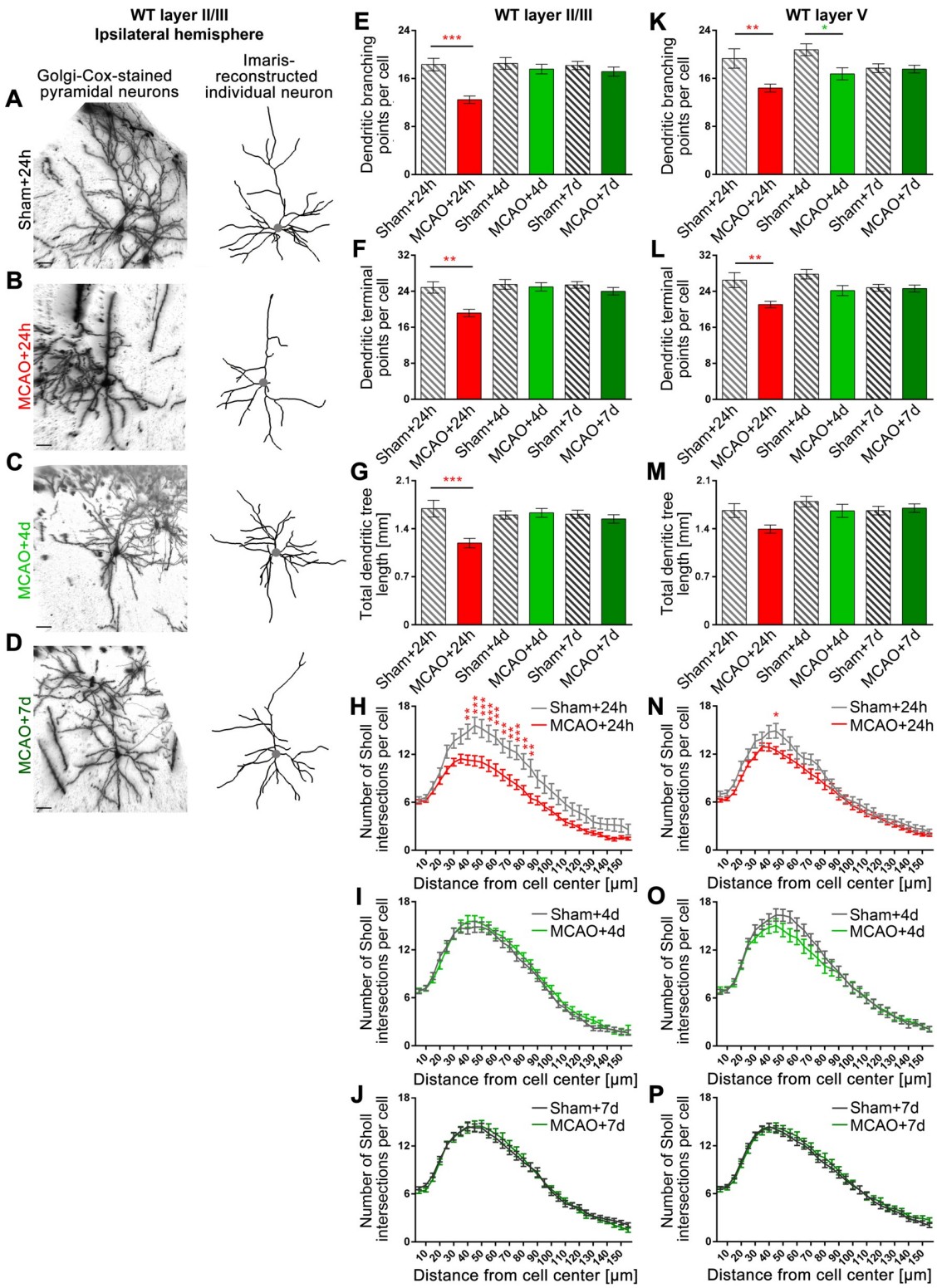

**Fig 5. Dendritic arborization defects caused by MCAO in M1 are completely reverted during the days after the ischemic event in both layer II/III and layer V neurons of the ipsilateral hemisphere. A–D**, Representative images of Golgi–Cox-stained and Imaris-reconstructed ipsilateral layer II/III pyramidal neurons of M1 of WT mice subjected to sham surgery and of WT mice subjected to 30-minute MCAO at 24 hours, 4 days and 7 days of reperfusion time. The positions of the cell bodies are marked by a gray dot. Scale bars, 30 μm. **E–J**, Quantitative determinations of dendritic arborization parameters of layer II/III neurons. Note that

dendritic branching points (**E**), dendritic terminal points (**F**), total dendritic tree length (**G**), and Sholl intersections (**H–J**) were all decreased upon MCAO after 24 hours but were fully recovered to values indistinguishable from the corresponding sham controls after 4 days and 7 days reperfusion, respectively. **K–P**, Quantitative analyses of dendritic arborization parameters of layer V neurons at 24 hours, 4 days, and 7 days reperfusion. Ipsilateral M1 tissue of 3 mice of an age of 3–4 months were analyzed for the sham+24h, sham+4d, and MCAO+4d groups and 6 mice for the sham+7d, MCAO+24h, and MCAO+7d groups, as 2 independent animal cohorts were evaluated and included (for initial MCAO+24h data from only 3 mice, e.g., see **Fig 4**) (for contralateral data, see **S5 Fig**). Layer II/III: $n_{Sham+24h}$ = 18; $n_{MCAO+24h}$ = 43; $n_{Sham+4d}$ = 35; $n_{MCAO+4d}$ = 32; $n_{Sham+7d}$ = 63; $n_{MCAO+7d}$ = 47 neurons. Layer V: $n_{Sham+24h}$ = 22; $n_{MCAO+24h}$ = 51; $n_{Sham+4d}$ = 27; $n_{MCAO+4d}$ = 29; $n_{Sham+7d}$ = 61; $n_{MCAO+7d}$ = 51 neurons. Data represent mean ± SEM. Statistical significances calculated between sham and MCAO of the corresponding reperfusion time using 2-way ANOVA with Sidak posttest are shown. $^{*}P < 0.05$; $^{**}P < 0.01$; $^{***}P < 0.001$. The numerical data underlying this figure can be found in **S5 Data**. MCAO, middle cerebral artery occlusion; WT, wild-type.

play some important role in shaping cortical neurons and whether furthermore Cobl may not only do so in developing neurons in culture but also in adult neurons in the cortex of mice with a demand for remapping neuronal circuits subsequent to stroke.

Addressing the first hypothesis, we overexpressed Cobl in developing rat cortical neurons. GFP-Cobl overexpression from DIV4 to DIV6 clearly resulted in increased dendritic arborization when compared to GFP control. All parameters affected upon MCAO in mice were elevated significantly (**Fig 6A–6F**).

In line with these results, Cobl loss-of-function experiments in developing primary rat cortical neurons unveiled that Cobl does not only have the ability to modulate the dendritic trees of cortical neurons but also is critical for this process during dendritic arbor developement (**Fig 6G–6I**). Dendritic branching points, terminal points, and the total dendritic length were significantly reduced and also Sholl analyses highlighted a loss of dendritic complexity when Cobl was lacking (**Fig 6J–6M**). All of these Cobl loss-of-function phenotypes were specific, as all of them could be rescued by reexpression of RNA interference (RNAi)-insensitive Cobl in the cultured cortical neurons (**Fig 6I–6M**).

Taken together, these 2 different experimental lines clearly proved the first hypothesis and unveiled that the actin nucleator Cobl played an important role in the dendritic arborization of cortical neurons.

## *Cobl* KO completely ablates the ischemic stroke–induced dendritic regrowth processes, which allow for restoration of proper dendritic arborization after MCAO

The second hypothesis, that, as reflected by the identified modulations of Cobl levels acutely following ischemia and during stroke recovery, the actin nucleator Cobl may be a crucial player in the repair of the stroke-induced dendritic arbor defects in the penumbra within the observed narrow time window between day 1 and day 4 after MCAO was addressed by subjecting *Cobl* KO mice [37] to comparative and time-resolved MCAO studies evaluating the different dendritic parameters in a fully blinded manner. Similar to WT mice (**Fig 5**), the *Cobl* KO mice also showed a reduction of dendritic branching points, terminal points, and total dendritic tree length 24 hours after MCAO. At 24 hours after MCAO, the reduction in each parameter was about 25% in *Cobl* KO mice (**Fig 7E–7G and 7K–7M**). These MCAO-induced declines were about as strong as detected in WT animals (**Fig 5**). Deviations in Sholl analyses of dendritic arbor complexity of *Cobl* KO neurons very well reflected the reductions in dendritic branching points, terminal points, and tree extension (**Fig 7H and 7N**). In general, all individual defects in *Cobl* KO mice were about equally strong in layer II/III and in layer V 24 hours after MCAO (**Fig 7E–7G and 7K–7M**).

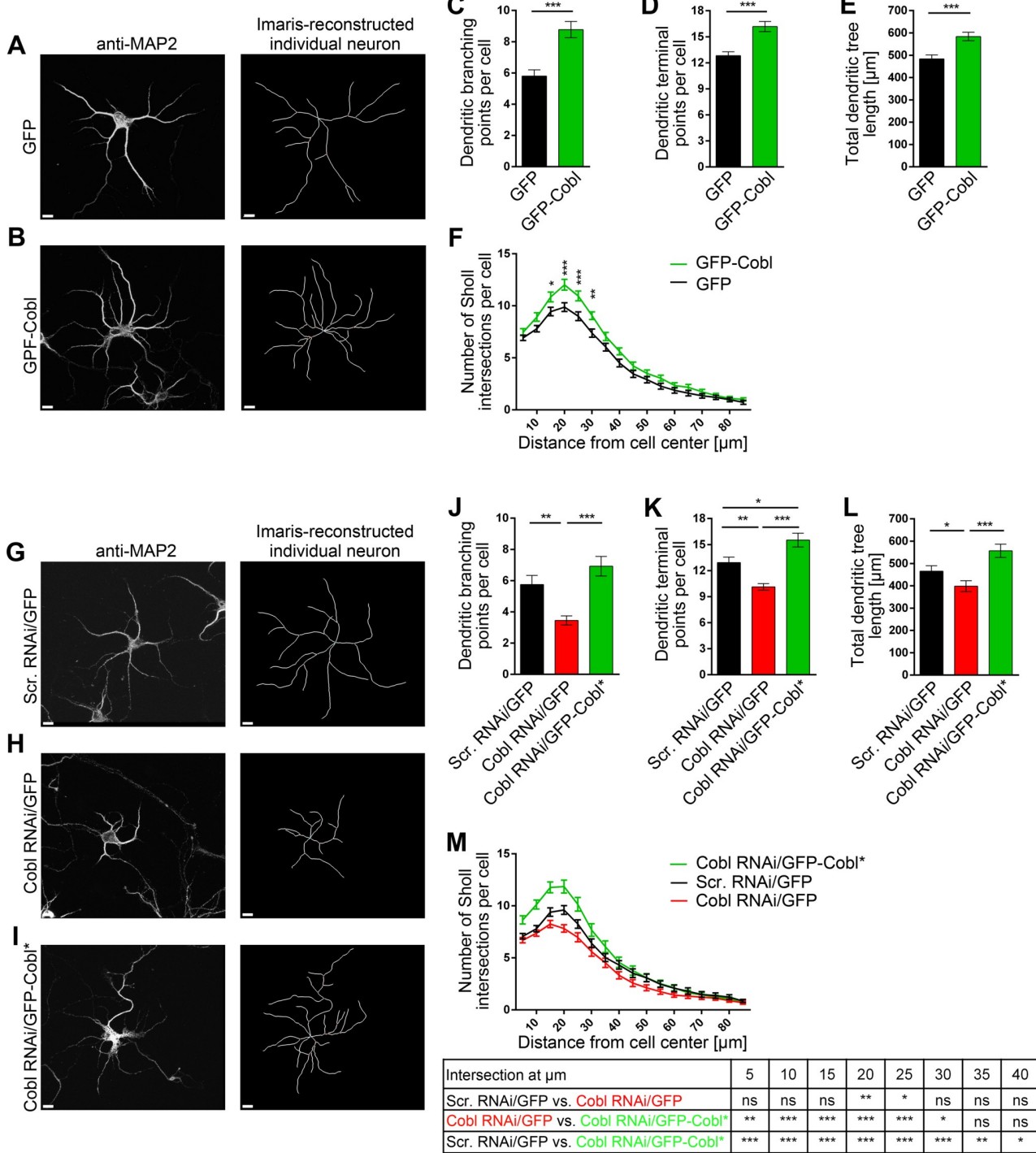

**Fig 6. The actin nucleator Cobl promotes and is critical for dendritic arborization during early development of cultured primary rat cortical neurons. A, B**, Representative MIPs of anti-MAP2-immunostained (left panels) and Imaris-reconstructed primary rat cortical neurons (right panels) that were transfected at DIV4 with either GFP (**A**) and GFP-Cobl (**B**), respectively, and fixed 40 hours later. Scale bars, 10 μm. **C–F**, Quantitative determinations of dendritic arborization parameters unveiling clear Cobl gain-of-function phenotypes. Note that dendritic branching points (**C**), dendritic terminal points (**D**), total dendritic tree length (**E**), and Sholl intersections (**F**) were all increased upon Cobl overexpression, respectively. **G–I**, Representative MIPs of anti-MAP2-immunostained (left panels) and Imaris-reconstructed primary rat cortical neurons (right panels) that were transfected at DIV4 with plasmids encoding for either GFP-reported scrambled RNAi (Scr. RNAi/GFP) (**G**), Cobl RNAi/GFP (**H**) and Cobl RNAi together with silently mutated Cobl rendered insensitive again Cobl RNAi (Cobl RNAi/GFP-Cobl*) (**I**), respectively, and fixed 40 hours later. Scale bars, 10 μm. **J–M**, Quantitative determinations of dendritic arborization parameters unveiling clear and specific Cobl loss-of-function phenotypes.

Note that dendritic branching points (**J**), dendritic terminal points (**K**), total dendritic tree length (**L**), and Sholl intersections (**M**) were all decreased upon Cobl RNAi and were rescued to levels clearly statistically different from Cobl RNAi by reexpressing GFP-Cobl*. $n_{GFP}$ = 40; $n_{GFP-Cobl}$ = 40; $n_{Scr.}$ $_{RNAi/GFP}$ = 40; $n_{Cobl RNAi/GFP}$ = 40; $n_{Cobl RNAi/GFP-Cobl*}$ = 40 individual transfected neurons from 2 independent preparations of cortical neurons. Data, mean ± SEM. Statistical significances were calculated using Student $t$ test (**C–E**), 1-way ANOVA with Tukey posttest (**J–L**), and 2-way ANOVA with Sidak posttest for Sholl analysis (**F, M**). *$P$ < 0.05; **$P$ < 0.01; ***$P$ < 0.001. The numerical data underlying this figure can be found in **S6 Data**. MIP, maximum intensity projection; RNAi, RNA interference.

Again similar to WT mice, the *Cobl* KO mice did not also show any significant MCAO-induced neuronal morphology changes on the contralateral side (**S6 Fig**), but MCAO-induced changes were restricted to the ipsilateral side (**Fig 7**).

By evaluating the restoration of proper dendritic arbor complexity after MCAO at the ipsilateral side, it became obvious that the repair of dendritic arborization subsequent to MCAO was grossly impaired in *Cobl* KO mice. All 4 quantified dendritic defects remained fully present 4 days after MCAO. Neither dendritic branching points (**Fig 7E and 7K**), nor dendritic terminal points (**Fig 7F and 7L**) nor dendritic tree length (**Fig 7G and 7M**) nor dendritic complexity, as visualized by Sholl analyses (**Fig 7I and 7O**), were restored to proper levels in *Cobl* KO animals. This phenotype became obvious by the comparison to sham-treated animals (**Fig 7**) but also by comparing the absolute numbers of all parameters with those of WT animals (**Fig 7** versus **Fig 5**).

At day 4, this complete lack of repair of MCAO-induced defects was observed in both layer II/III neurons (**Fig 7C, 7E–7G and 7I**) and in layer V neurons of *Cobl* KO mice (**Fig 7K–7M and 7O**). The lack of repair at day 4 also occurred irrespective of the dendritic orientation, as MCAO-induced defects in *Cobl* KO neurons were observed in both the apical and the basal dendritic arbor (**S7 Fig**).

Even at day 7 after MCAO and again in strong contrast to WT animals, *Cobl* KO mice still showed massive impairments in dendritic arborization (**Fig 7D, 7E–7G, 7J and 7K–7M**). Only Sholl analyses of layer V neurons failed to visualize the differences in dendritic complexity between sham- and MCAO-treated *Cobl* KO mice in a manner supported by statistical significances (**Fig 7P**). Yet, all other 7 quantitative examinations still clearly demonstrated the lack of any dendritic arbor restoration in *Cobl* KO mice even a full week after MCAO (**Fig 7E–7P**). The dendritic branching points, the terminal points, and the summarized length of the dendritic arbor remained significantly reduced when compared to sham-treated animals at day 7. The defects at day 7 thereby still remained similar to the defects observed at 24 hours and 4 days (**Fig 7E–7P**). Taken together, the levels of the actin nucleator Cobl are responsive to MCAO-induced brain damage, and Cobl is critically needed for an acute process of dendritic regrowth and branching that is triggered by ischemic stroke conditions and is powerful enough to fully repair the strong defects in dendritic arborization, which we observed as consequences of ischemic stroke in the cortex.

## Discussion

Regaining of neurophysiological functions in peri-infarct areas (the penumbra) and remapping processes in functionally related cortical tissues are thought to represent important aspects underlying recovery from stroke in human patients. However, such processes are far from being understood at the cellular and molecular level.

We demonstrated that ischemic stroke in mice leads to a strong decline in the entire dendritic arborization of neurons in the penumbra. By carefully comparing different reperfusion times with each other and by analyzing sets of corresponding contralateral controls, we show that the dendritic arborization loss is an ipsilateral phenomenon and demonstrate a subsequent ipsilateral process of dendritic arbor repair. In the cortex of mice subjected to 30-minute

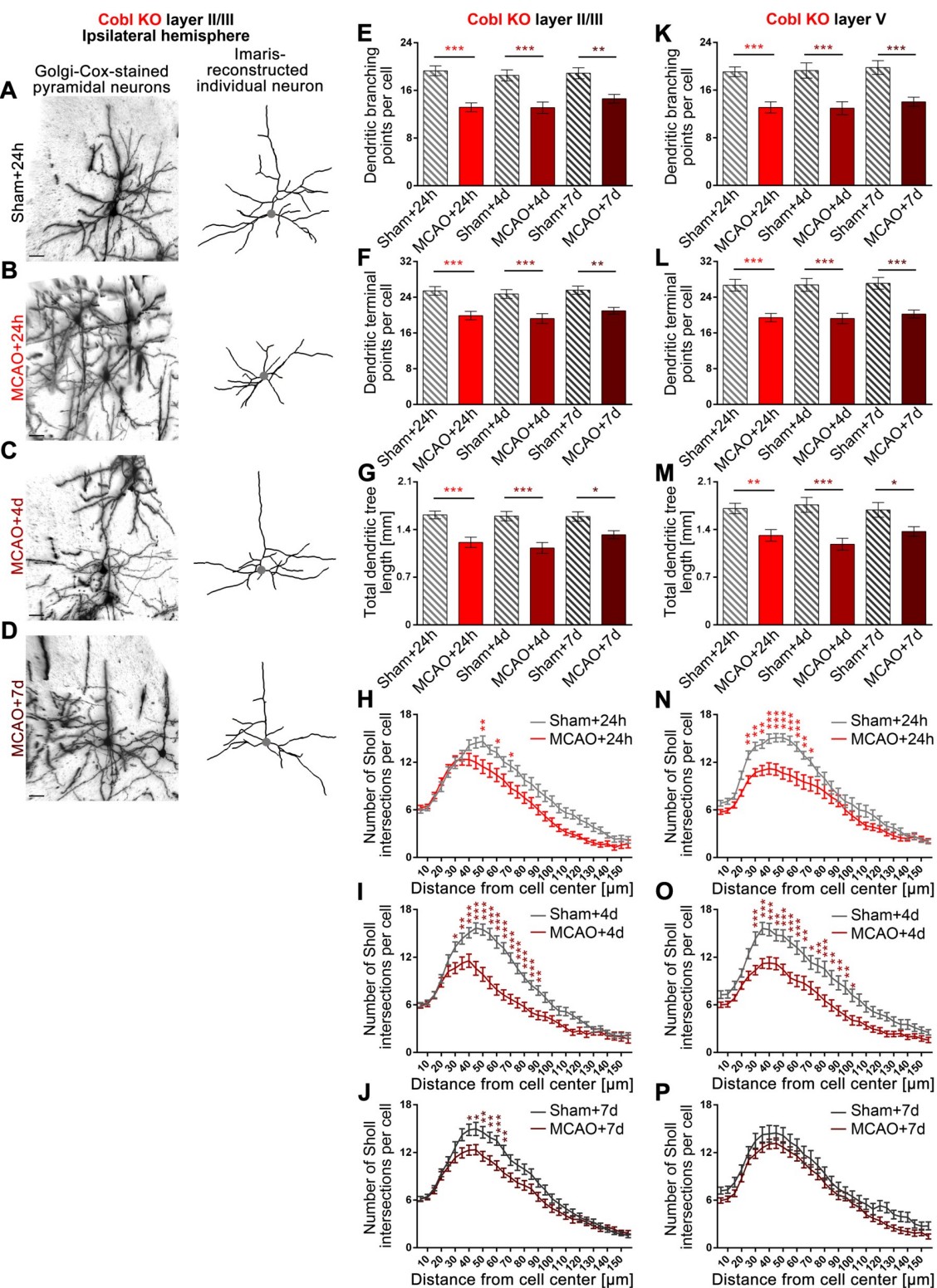

**Fig 7. _Cobl_ KO mice fail to show repair of the dendritic arborization defects caused by MCAO in both layer II/III and layer V neurons of the ipsilateral hemisphere. A–D**, Representative images of Golgi–Cox-stained and Imaris-reconstructed ipsilateral layer II/III pyramidal neurons of M1 of _Cobl_ KO mice subjected to sham surgery and of _Cobl_ KO mice subjected to 30-minute MCAO at 24 hours, 4 days, and 7 days of reperfusion time. The position of the cell bodies are marked by a gray dot. Scale bars, 30 μm. **E–J**, Quantitative determinations of dendritic arborization parameters of layer II/III neurons. Note that dendritic branching points (**E**),

dendritic terminal points (**F**), total dendritic tree length (**G**), and Sholl intersections (**H-J**) were all decreased at 24 hours after MCAO and that, in contrast to WT animals (**Fig 5**), none of these phenotypical parameters recovered during the days after MCAO in *Cobl* KO mice. Instead, all dendritic parameters remained strongly suppressed even after 4 days and 7 days of reperfusion. **K–P**, Quantitative analyses of dendritic arborization parameters of layer V neurons in M1 at 24 hours, 4 days, and 7 days reperfusion showing a similar lack of dendritic recovery from ischemic stroke upon *Cobl* KO. For each MCAO and sham group, 3 mice of an age of 3 to 4 months were analyzed (for contralateral data, see **S6 Fig**). Layer II/III: $n_{Sham+24h} = 21$; $n_{MCAO+24h} = 35$; $n_{Sham+4d} = 30$; $n_{MCAO+4d} = 29$; $n_{Sham+7d} = 26$; $n_{MCAO+7d} = 35$ neurons. Layer V: $n_{Sham+24h} = 25$, $n_{MCAO+24h} = 31$, $n_{Sham+4d} = 21$, $n_{MCAO+4d} = 25$, $n_{Sham+7d} = 19$, $n_{MCAO+7d} = 30$ neurons. Data represent mean ± SEM. Statistical significances calculated between sham and MCAO of the corresponding reperfusion time using 2-way ANOVA with Sidak posttest are shown. $^*P < 0.05$; $^{**}P < 0.01$; $^{***}P < 0.001$. The numerical data underlying this figure can be found in **S7 Data**. KO, knockout; MCAO, middle cerebral artery occlusion.

MCAO-induced ischemic stroke in the striatum, this repair process showed a net growth that was able to fully compensate for the massive loss of dendritic extension and complexity observed at the first day after ischemic stroke. Importantly, with the actin nucleator Cobl, we identified a cytoskeletal driving force for this recovery. Dendritic arbor repair was completely absent in *Cobl* KO mice subjected to ischemic stroke. Neurons in M1 of *Cobl* KO mice subjected to MCAO remained at the state of severe dendritic impairment.

During the repair in WT animals, the observed growth represented an increase of the dendritic arbor by one-third above 24 hours poststroke levels. Such significant changes in dendritic arborization in adult animals are quite stunning, as the dendritic arbor, in contrast to dendritic spines, which are known to at least retain some minor dynamics in adults, is established to be rather resilient under normal conditions [14]. Recent observations in rats subjected to transient MCAO [27], however, are fully in line with the drastic changes we observed in mice. Although not fully comparable to our studies, as a stronger MCAO was used (90 minutes), ischemic stroke was reported to result in significant declines in the mean length and branching of neurons in the ipsilateral somatosensory cortex at day 3—a defect not observed anymore at the 21-day time point studied for comparison [27]. Also, 3 repetitive vascular endothelial growth factor D (VEGFD) applications over 48 hours leading to a net dendritic growth of more than a third within 5 days post-MCAO [28] are well in line with the observed net dendritic growth rates in our study.

In this respect, it is also very interesting that post-photothromic ischemia in vivo imaging showed a massive remodeling of dendrites in the somatosensory cortex at 2 weeks after stroke but no net growth [20]. This may suggest that the massive induction of net dendritic growth and the branch induction we observed when studying much earlier time points (day 2 to 4) actually still continue after reaching net recovery at day 4 but that later stages of cortical remapping are marked by counteraction by dendritic pruning processes operating in parallel.

Such a biphasic repair process would also explain apparently different findings concerning the effects within the dendritic arbor. During the day 2 to 4 time window of Cobl-dependent acute ischemic stroke–induced repair, all aspects of dendritic arborization were promoted with equal efficiency. The induced repair also did not discriminate between apical and basal dendrites or between the proximal and distal arbor but promoted neuronal morphology in all dendritic areas equally well (this study). By contrast, later dendritic dynamics did not result in any net growth anymore, as dendritic pruning processes operating in parallel lead to higher net growth away from the infarct area and to a net loss of dendrites oriented toward the infarct area [20]. Together, such a biphasic response would very effectively remap neuronal circuits in the neighborhood of infarct areas.

The observed readdition of on average 5 to 6 dendritic branching points and of about 400 μm of dendritic arbor per neuron in peri-infarct areas in a time window of repair, which opens at or after day 2 after ischemic stroke, obviously provides a powerful mechanism for not only reconnecting cells that anyway are close enough for synapse formation but also allows for

making connections over larger ranges. Consistently, cortical remapping is an important mechanism in recovery after ischemic stroke [47].

Neurons in the penumbra of *Cobl* KO mice subjected to ischemic stroke failed to show any postischemic repair. Even 4 and 7 days after MCAO, all quantitative parameters of the dendritic arbor of both layer II/III and layer V neurons in M1 still remained severely affected irrespective of whether the complete dendritic arbor was analyzed or apical and basal dendrites were evaluated separately, whereas WT mice accomplished a full dendritic regrowth in a time frame of about 3 days following the most severe defects at 24 hours poststroke.

What is so special about the actin nucleator Cobl that it is employed in repair processes aiming at regaining brain functions after ischemic stroke in the cortex? Cobl is a powerful actin nucleator that can even work with very low levels of ATP-loaded G-actin, as demonstrated in reconstitutions of actin nucleation with as little as 2 μM actin [30]. Unlike, e.g., the Arp2/3 complex, which represents a major actin nucleator for many basic functions of mammalian cells [48], Cobl is an evolutionary relatively young nucleator with rather specialized functions in a distinct set of specialized cells. These include embryonic hippocampal neurons [30], early postnatal Purkinje cells in the cerebellum [32] and, as demonstrated here in adult mice, also layer II/III and V pyramidal neurons in peri-infarct areas after ischemic stroke. In addition to neurons, these specialized cells also include early postnatal outer hair cells in the cochlea [37] and short-lived enterocytes in the small intestine in mice [49] as well as motile cilia-carrying cells in the Kupffer's vesicle [50] and sensory stereocilia- and kinocilia-bearing neuromasts of the lateral line organ [51] in fish larvae. In both outer hair cells in the postnatal cochlea and in enterocytes of the small intestine, Cobl seems to be responsible for a specialized set of actin filaments [37,49]. At nascent dendritic branch initiation sites in embryonic neurons, Cobl and F-actin accumulated shortly before and during the branch induction step that breaks the cylindrical symmetry of the respective dendrite segment [36]. With such specialized cytoskeletal functions, Cobl can thus likely be specifically applied in distinct cellular functions without affecting basic functions of the actin cytoskeleton of a given cell.

A prerequisite for such a molecular scenario of course would be that Cobl's activity and membrane targeting can be regulated in a distinct manner. Indeed, Cobl functions require arginine methylation of one of its 3 G-actin binding WH2 domains by the arginine methyltransferase PRMT2 [35]. Furthermore, Cobl's N terminal membrane-associating domain (the Cobl Homology domain) interacts with syndapin I in a $Ca^{2+}$/CaM-regulated manner in early neuromorphogenesis [36] and links Cobl to its functional partner Cobl-like [34]. Strikingly, also Cobl's actin nucleating, WH2 domain-containing C-terminal part was found to be controlled by association of $Ca^{2+}$-activated CaM [36].

As unveiled here, $Ca^{2+}$ can also have very detrimental effects on the fate of Cobl and can lead to proteolytic destruction of Cobl. These yin and yang effects of $Ca^{2+}$ on the functions of Cobl are reminiscent of the concept of neurohormesis, in which neurotransmitter stimuli at low and medium intensity are beneficial and lead to functional adaptation, such as synaptic plasticity, whereas high loads of neurotransmitter lead to excitotoxicity [52,53]. NMDA receptor–mediated $Ca^{2+}$ influx is an important trigger in glutamate-induced neuroexcitotoxicity [40,54,55]. Consistently, we found that the proteolytic degradation of the actin nucleator Cobl in cortical neurons (i) could be induced by incubation with glutamate; (ii) was mediated by $Ca^{2+}$ and by calpain; and (iii) required the opening of NR2B subunit-containing NMDA receptors.

In line with an acute demand of Cobl for dendritic arbor repair after ischemic stroke, Cobl protein levels lost upon the $Ca^{2+}$/calpain-mediated proteolysis were restored to normal levels already 24 hours after MCAO. The discovery of increasing *Cobl* mRNA levels during the hours prior to this restoration of Cobl protein levels strongly suggests that this replenishment of Cobl levels is achieved by the observed rise in mRNA level. In line with the decline of Cobl protein

levels in M1, the transient elevation of *Cobl* mRNA levels could be also detected in the M1, i.e., in the penumbra. While nothing is known yet about how the expression of Cobl is controlled, it seems likely that the mRNA increase will be linked to signaling cascades triggered by high glutamate, such as massive increases in cytosolic and thereby also nuclear $Ca^{2+}$ concentrations.

The $Ca^{2+}$/calpain-mediated destructions of the important postsynaptic scaffold protein PSD95, of spectin/fodrin and of the NR2B subunit of NMDA-type glutamate receptors during the first hours after ischemic stroke received great attention, as they can be considered as being linked to the loss of synaptic connections [11–13]. Surprisingly, none of these components showed any signs of rapid counteraction by increase of mRNA levels. Thus, the transient increase of *Cobl* mRNA coinciding with subsequent restoration of normal Cobl protein levels and initiation of dendritic arbor repair seemed to be a unique response to ischemic conditions. The changes in *Cobl* mRNA levels and the subsequent restoration of Cobl protein levels we observed furthermore are (i) consistent with each other; and (ii) in line with the urgent requirement of Cobl for dendritic arbor repair after ischemic stroke.

Taken together, our study unveiled that ischemic stroke causes damages in dendritic arborization in peri-infarct areas, which then are repaired by processes of dendritic regrowth relying on the actin nucleator Cobl, whose levels upon ischemic stroke first are negatively affected but then are rapidly restored. The excitotoxicity-induced degradation of Cobl was caused by glutamate and NMDA receptor–mediated $Ca^{2+}$ influx and calpain-mediated proteolysis. Subsequent to a rapid restoration of normal Cobl levels, the actin nucleator then powered dendritic repair during a time window opening between day 2 to 4 after ischemic stroke. We show that Cobl-dependent poststroke dendritic arbor regrowth is a powerful cell biological process that adds to the well-known structural plasticity of postsynapses/dendritic spines. Importantly, Cobl-dependent poststroke dendritic arbor regrowth represents a much more long-range mechanism of poststroke repair inside of peri-infarct areas.

The high conservation of Cobl between mice and man suggests that related Cobl-dependent processes of repair after ischemic stroke may also exist in human patients and would be worthwhile to exploit in the identified poststroke time window.

## Material and methods

### Experimental model and subject details

***E. coli* strain XL10 gold.**   *E. coli* strain XL10 Gold (StrataGene, La Jolla, USA/Agilent, Santa Clara, USA) (genomic information: endA1 glnV44 recA1 thi-1 gyrA96 relA1 lac Hte Δ(mcrA)183 Δ(mcrCB-hsdSMR-mrr)173 tetR F'[proAB lacIqZΔM15 Tn10(TetR Amy CmR)]) was used for amplifications of DNA plasmids. The strain was grown in LB medium (Carl Roth, Karlsruhe, Germany) or on LB-agar (Carl Roth) with or without the respective antibiotics the strain carries resistances for (10 µg/ml tetracycline and 30 µg/ml chloramphenicol) or obtained resistances for (by transformation with plasmids) (usually either 100 µg/ml ampiciline or 25 µg/ml kanamycine). The strain was usually either grown at room temperature (RT) or at 37˚C.

**HEK293 cells.**   Human Embryonic Kidney 293 (HEK293) cells were cultured in Dulbecco's Modified Eagle Medium supplemented with 10% (v/v) fetal bovine serum, 0.1 mM gentamicin, penicillin (100 units/ml), and streptomycin (100 µg/ml) in an cell incubator (5% $CO_2$, 37˚C, 90% humidity) and split every 3 to 4 days.

The sex of the cells is not specified. The original publication reports that "primary or early passage secondary human embryonic kidney (HEK) cells prepared by standard techniques" were used [56].

**Primary rat neurons.** Rat cortical neurons (rat strain, Wistar) were prepared as described [57] and cultured until DIV4 and DIV15, respectively.

**Mice.** All studies were performed with 3 to 4 months old male WT and *Cobl* KO mice, respectively. *Cobl* KO mice were described previously [37]. They were generated via excision of exon 11 of the *Cobl* gene using the Cre/loxP system and subjected to speed congenics [37]. For the current study, *Cobl* KO mice were backcrossed to C57BL/6J until C57BL/6J::129/SvJ (97.375::2.625) was reached (infarct volumetry) (B6.129/Sv-Cobl[tm1BQ]). Neuromorphometric analyses were mostly done with even further backcrossed *Cobl* KO animals (up to 99.918% C57BL/6J). Despite the high C57BL/6J genetic backgrounds, comparisons with WT animals were done with WT littermates of heterozygous breedings of *Cobl* KO mice in both infarct volumetry and morphometric analyses of stroke damage and repair processes.

Additionally, tissue material for qPCRs and western blot analyses as well as for brain sections for anti-MAP2 and Golgi–Cox stainings were also taken from WT C57BL/6J mice when MCAO conditions were examined and compared in different WT conditions.

## Methods

### Breeding, housing, and handling of mice

Mice were kept in a specific pathogen-free environment at RT (22°C) with the humidity of 68% under 14-hour light/10-hour dark conditions with ad libitum access to food and water. Mice were regularly tested for parasites and other pathogens using sentinel mice held under identical conditions.

All animal care and experimental procedures were performed in accordance with EU animal welfare protection laws and regulations and were approved by a licensing committee from the local government (Landesamt für Verbraucherschutz, Bad Langensalza, Thuringia, Germany). The generation of *Cobl* KO mice was approved by the permission number 02-011/10. Breeding and housing was approved by the permission number UKJ-17-021. MCAO experiments with WT mice were approved by permission numbers 02-024-15 and 02-057-14, respectively. The MCAO experiments conducted with WT and *Cobl* KO littermates from heterozygous breedings of *Cobl* KO mice were approved by the permission number UKJ-18-030 (Landesamt für Verbraucherschutz, Bad Langensalza, Thuringia, Germany).

### MCAO

MCAO can be experimentally achieved in mice and rats by temporarily occluding the common carotid artery (CCA), introducing a suture directly into the internal carotid artery (ICA) and advancing it until it interrupts the blood supply to the MCA [7,58]. MCAO therefore is one of the most frequently used experimental ischemic stroke model systems. Thirty minutes of ischemia causes neuronal cell death limited to the striatum. MCAO was induced as previously described [59]. In brief, mice were treated with Melosus and anesthetized with 2.5% (v/v) isoflurane in $N_2O:O_2$ (3:1). Through a middle neck incision, the right CCA, the external carotid artery (ECA), and the ICA were carefully dissected from surrounding nerves and fascia. A 7–0 nylon monofilament (70SPRe, Doccol, Sharon, USA) coated with silicone rubber on the tip (0.20 ± 0.01 mm diameter) was introduced into the ICA through an incision of the right CCA up to the circle of Willis. In this position, the suture occluded MCA. During the MCA occlusion, body temperature was maintained at physiological level using a heating pad. After 30 minutes of occlusion, the suture was withdrawn to restore the blood flow and allow for reperfusion.

Sham animals underwent anesthesia and the same surgical procedures except the occlusion of MCA.

## Golgi–Cox staining, cryosectioning, and infarct validation

Golgi–Cox staining (Golgi–Cox impregnation) for studying the morphology of individual neurons in the brain [60] was done with brain material from adult (3 to 4 months old), male mice that were subjected to either sham or 30-minute MCAO and killed after different reperfusion times. The brains were quickly removed and separated into 2 parts using a Precision Brain Slicer (Braintree Scientific, Sarasota, USA) at bregma +0.8 mm using the Allen Brain Atlas as reference. The anterior part was used for the infarct validation by anti-MAP2 staining (see below), and the posterior part was used for Golgi–Cox staining using FD Rapid GolgiStain kits (FD NeuroTechnologies, Columbia, USA), essentially following a protocol described previously [61,62]. In brief, the posterior brain part was immersed in solution A+B of the FD Rapid GolgiStain kit and kept in the dark at RT for 21 days. Brains were then incubated in solution C of the kit for another 5 days (RT; dark). The brains were then dipped very slowly into dry ice-precooled isopentane and stored at −80°C.

Cryosectioning was performed at −28°C using a Leica CM 3050 S (Leica Biosystems Nussloch, Nußloch, Germany). Coronal sections of 100-μm thickness were cut and transferred onto small drops of solution C of the FD Rapid GolgiStain kit on gelatine-coated slides. After drying overnight (RT; dark), the sections were rinsed twice with water and stained for 1 minute in the staining solution of the kit. After rinsing twice with water, sections were dehydrated in an ascending ethanol series (50%, 75%, 95%, 4 times 99% (v/v)) and cleared 3 times in xylene. Afterward, the Golgi–Cox-stained tissue sections were mounted onto coverslips with Roti-Histokitt (Carl Roth) and stored in the dark.

The anterior part of the brains cut at bregma +0.8 mm was used for anti-MAP2 immunostainings. For this purpose, free-floating sections were pretreated with 0.24% (v/v) $H_2O_2$ in Tris-buffered saline (83.8 mM Tris-HCl; 16 mM Tris-Ultra (Base) pH 7.4, 154 mM NaCl, 0.2% (v/v) Triton X-100) (TBS-T) for 30 minutes and blocked with TBS-T containing 3% (v/v) normal donkey serum (NDS) (GeneTex, California, USA), 2% (w/v) bovine serum albumin (Sigma-Aldrich, St. Louis, USA), 3% (w/v) milk powder, and Fab fragments of donkey anti-mouse IgG (1:200) (Jackson ImmunoResearch, Ely, UK).

Sections were incubated at 4°C overnight with monoclonal antibody against mouse MAP2 (Sigma-Aldrich) (1:1,000 in TBS-T with 3% (v/v) NDS) and then with a biotinylated secondary antibody (Jackson ImmunoResearch) (1:500 in TBS-T with 3% (v/v) NDS) after 3 times washing with TBS-T and 1 time with 3% (v/v) NDS. In the end, the sections were processed with Vectastain Elite ABC HRP kits (Vector Laboratories, Burlingame, USA) for 1 hour, and immunosignals were visualized using 3,3′-diaminobenzidine reactions. Brains of MCAO-treated mice displaying an anti-MAP2-negative infarct area in the ipsilateral hemisphere were then subjected to the neuromorphological analysis and compared to brains of sham-treated animals processed the same way. Blinding was done by using multiple digit animal numbers not indicative of genotype, MCAO or sham treatment, or the different reperfusion times and by using Antrenamer2 blinding software.

## Quantitative analyses of neuronal morphology in the M1

Image acquisitions for neuronal morphology analyses were done from 3 to 5 independent coronal sections containing the M1 area per mouse (bregma +0.8 to 0 mm). Layer II/III and layer V pyramidal neurons in M1 were identified by their distance from pia and their distinct morphologies. Z-stacks of Golgi–Cox-stained neurons were recorded at 20× magnification (1-μm intervals), using a Zeiss AxioObserver equipped with a Zeiss Plan-Apochromat 20×/0.5 objective and an AxioCam MRm CCD camera (Zeiss, Jena, Germany). Digital images were acquired by AxioVision software (Zeiss).

Overview images of mouse brain sections were generated by stitching images recorded with a 5×/0.16 objective together using the Tiles mode of the microscope and the automatic stitching of the Zen Software (Zeiss) with an overlap of 10%.

Three-dimensional reconstructions of neuronal morphologies were performed with Imaris 8.0 software (Bitplane, Zürich, Switzerland) using the filament tracer set with shortest distance algorithm (diameter: 2 to 16 μm). The images were baseline- and background-subtracted as described before [62]. As in previous software-based quantitative determinations of dendritic arbor complexity evaluations [63], the following parameters were analyzed: the number of Sholl intersections, the number of dendritic branching points, the numbers of dendritic terminal points, and the total dendritic tree length.

The basal dendrites and their branches as well as the arbor originating from the apical dendrite were analyzed accordingly by dissecting the Imaris tracings of evaluated layer II/III WT and *Cobl* KO neurons.

The analyses were conducted in a blinded manner. Quantitative parameters were analyzed for data distribution and statistical significances with Prism 6 software (SCR_002798; Graph-Pad, San Diego, USA).

## Cobl gain- and loss-of-function studies in primary rat cortical neurons during early development

Rat cortical neurons were prepared and transfected with plasmids (GFP and GFP-Cobl for gain-of-function experiments; scrambled (scr.) RNAi/GFP (control) and Cobl RNAi/GFP [30] as well as Cobl RNAi/GFP-Cobl* (rescue; [32]) for loss-of-function experiments) at DIV4, as described [57]. Neurons were fixed 40 hours after transfection by 4% (w/v) paraformaldehyde (PFA) for 7 minutes, then permeabilized and blocked by blocking solution consisting of 10% (v/v) horse serum, 5% (w/v) bovine serum albumin, and 0.2% (v/v) Triton X-100 in PBS for 1 hour at RT. After the immunostaining with anti-MAP2 antibodies (secondary antibodies, Alexa Fluor 568-labeled donkey anti-mouse (Molecular Probes, Eugene, USA (A10037)), the transfected neurons from 2 independent coverslips per condition per assay were imaged using a Zeiss AxioObserver Z1 microscope/ApoTome.

Morphometric measurements were based on the MAP2 signals using Imaris 8.4.0 software. The detailed settings were the following: thinnest diameter and gap length: 2 μm, minimum segment length: 10 μm. The number of dendritic branching points, terminal points, total dendritic tree length, and Sholl intersections were analyzed as described above for Golgi-stained neurons in M1.

## Determination of infarct volumes

The volumes of infarct areas after 30-minute MCAO were determined after 7 days of reperfusion according to procedures described previously [64]. In brief, *Cobl* KO and WT mice were anesthetized by isoflurane and transcardial perfusion was performed with 4% (w/v) PFA in 0.2 M sodium phosphate buffer pH 7.4. Brains were then removed and postfixed in 4% (w/v) PFA for 5 hours, cryoprotected in 0.1 M sodium phosphate buffer (pH 7.4) containing 10% (w/v) sucrose overnight and transferred to 30% (w/v) sucrose in 0.1 M sodium phosphate buffer (pH 7.4) until saturation. After freezing, the brains were stored at −80˚C.

Coronal sections (40 μm) were sliced with a cryomicrotome (Thermo Fisher Scientific, Waltham, USA). The sections were maintained in anti-freeze buffer (0.05 M sodium phosphate buffer (pH 7.4) containing 3 mM NaN$_3$, 832 mM glucose and 30% (v/v) ethylene glycol) at −20˚C until use. Free-floating sections were subjected to anti-MAP2 immunostainings as described above.

The anti-MAP2 stained brain sections were imaged on a light table using a digital CCD camera (Hamamatsu Photonics, Hamamatsu, Japan) and the sizes of the infarct area as well as of each hemisphere and of the total brain section were measured by Scion Image software (Scion, Frederick, USA). The area of the infarct was traced and quantified on every twelfth section. Infarct volumes were calculated as percentage of the total brain.

### Brain tissue preparations for subsequent qPCR and western blot analyses

Mice subjected to 30-minute MCAO were decapitated under deep anesthesia at 2 hours, 3 hours, 6 hours, 12 hours, 24 hours, and 48 hours reperfusion time, respectively. Brains were removed, and tissue samples for expression analyses were prepared as described previously [59]. In brief, brains were cut into 3 coronal segments using a Precision Brain Slicer (Braintree Scientific): from +2.8 to +0.8 mm to bregma (rostral segment), from +0.8 to −1.2 mm to bregma (middle segment), and from −1.2 to −3.2 mm to bregma (caudal segment). The ipsilateral part (ischemic hemisphere) and the contralateral part (nonischemic hemisphere) of the middle brain segment were separated and quick frozen in liquid nitrogen and stored at −80°C for later processing.

In further animal experiments (after 6 hours, 12 hours, and 24 hours reperfusion time), the middle brain segment was dissected further and the primary motor cortex M1 was isolated and snap-frozen for subsequent analyses of exclusively penumbral ipsilateral and contralateral brain areas.

### qPCR analyses

Slices from rostral and caudal segments that were adjacent to the middle segment were subjected to anti-MAP2 staining for infarct validation to select mice with similar lesion size for mRNA quantifications, as described before [55].

In brief, the ipsilateral and contralateral parts of the middle brain segment and M1 tissue samples of mice with comparable MCAO-induced lesions prepared as described above were homogenized in QIAzol Lysis Reagent (Qiagen, Hilden, Germany) followed by the addition of chloroform (Sigma-Aldrich) to separate the homogenate into a clear upper aqueous layer (containing RNA), an interphase, and a red lower organic layer (containing the DNA and proteins). RNA was precipitated from the aqueous layer with isopropanol. The precipitated RNA was washed with 75% (v/v) ethanol and then resuspended in pure $H_2O$ (Gibco, Waltham, USA). The quality and quantity of the RNA were measured by Nanodrop 2000 (Thermo Fisher Scientific).

After adjustment of RNA concentration, equal amounts of total RNA (0.5 μg) were transcribed to cDNA with RevertAid First Strand cDNA synthesis kit (Thermo Fisher Scientific). In detail, 0.6 μl oligo(dT)18 primer (stock, 100 μM), and 0.5 μl random hexamer primer (stock, 100 μM) were mixed and added to 5 μl RNA (100 ng/μl), and then incubated in a thermal cycler for 5 minutes at 65°C. Meanwhile, 2 μl 5× reaction buffer, 1 μl dNTP (10 mM), 0.5 μl RiboLock RNase inhibtor (20 U/μl) and 0.5 μl Revert Aid reverse transcriptase (200 U/μl) were mixed and then added to the preincubation mix. Transcription was done by incubations at 25°C (5 minutes), at 42°C (60 minutes), and finally at 70°C (5 minutes). The obtained cDNA was then diluted to the final concentration of 5 ng/μl and used for qPCR analyses.

qPCR was performed in a volume of 20 μl containing Brilliant III SYBR Green qPCR Master Mix (Agilent), cDNA (equivalent to 25 ng reversely transcribed RNA), and specific primers each at a final concentration of 500 nM. Primers used are listed in **Table 1**.

Amplification was performed using a Rotor-Gene 6000 (Qiagen) (cycle conditions: 3 minutes polymerase activation, 40 amplification cycles of 95°C for 10 seconds and 60°C for 15 seconds each).

**Table 1. qPCR primers used in the study.**

| Gene | Product (bp) | Accession number | Aligns to (bp) | Sequence 5′→3′ |
|---|---|---|---|---|
| *Calpain I* | 129 | NM_007600.3 | F: 1,137–1,156 | GGGGAGTTCTGGATGTCGTT |
| | | | R: 1,265–1,245 | GGTGCCCTCGTAAAATGTGGT |
| *CaM 1* | 99 | NM_009790.5 | F: 442–463 | GAAGTGGATGCTGATGGCAATG |
| | | | R: 540–519 | GCGGATCTCTTCTTCGCTATCT |
| *Cobl* | 134 | NM_172496.3 | F: 415–434 | GAACAGCACCTTGGACATCA |
| | | | R: 548–529 | CGTGGTGAGAAGGATTCAGG |
| *Cobl b* | 249 | NM_172496.3 | F: 685–705 | GGCTCCTGAGAAATCTGTACG |
| | | | R: 933–910 | CTAAACATTTCTCTTCTGTTGTCC |
| *Cobl c* | 135 | NM_172496.3 | F: 607–626 | GATTGGGTCCTTGAATGTGC |
| | | | R: 741–723 | ACAGCCTTCTGCGTTCTCA |
| *Gapdh* | 164 | NM_008084.2 | F: 1,092–1,112 | CAACAGCAACTCCCACTCTTC |
| | | | R: 1,255–1,234 | GGTCCAGGGTTTCTTACTCCTT |
| *Hmbs* | 106 | NM_013551.2 | F: 1,132–1,151 | GTTGGAATCACTGCCCGTAA |
| | | | R: 1,237–1,218 | GGATGTTCTTGGCTCCTTTG |
| *NR2B* | 150 | NM_008171.4 | F: 2,931–2,951 | AGGGGTGTAGATGATGCCTTG |
| | | | R: 3,080–3,061 | GCCCGTAGAAGCAAAGACCT |
| *PSD95* | 137 | NM_007864.3 | F: 361–379 | GATGAAGACACGCCCCCTC |
| | | | R: 497–478 | ATCTCCCCCTCTGTTCCGTT |
| *Spectrin 2* | 133 | NM_001177668.1 | F: 1,322–1,341 | CGCTTTCTTGCTGACTTCCG |
| | | | R: 1,454–1,435 | CACCCTTGTGCTCTTGATGC |
| *Tubb3* | 133 | NM_023279.2 | F: 291–311 | GCCTTTGGACACCTATTCAGG |
| | | | R: 423–404 | ACTCTTTCCGCACGACATCT |

qPCR, quantitative PCR.

Gene expression data were normalized to *glyceraldehyde 3-phosphate dehydrogenase* (*Gapdh*), *hydroxymethylbilane synthase* (*Hmbs*), and *tubulin beta 3 class III* (*Tubb3*) data, respectively (*Gapdh*, *Hmbs*, and *Tubb3* primers; see **Table 1**). Relative expression levels were calculated using the Pfaffl equation [65], $ratio = \frac{(E_{\text{target}})^{\Delta CP_{\text{target}}(control - sample)}}{(E_{\text{ref}})^{\Delta CP_{\text{ref}}(control - sample)}}$, with $E_{\text{target}}$ being the real-time PCR efficiency of target gene amplification, $E_{\text{ref}}$ being the real-time PCR efficiency of a reference gene transcript, $\Delta CP_{\text{target}}$ being the CP deviation of contra minus ipsi data of the target gene transcript and $\Delta CP_{\text{ref}}$ being the CP deviation of contra minus ipsi data of the reference gene.

## Protein isolation from mouse brain tissue for quantitative immunoblotting analyses

The ipsilateral part and the contralateral part, respectively, of the middle brain segment (+0.8 and −1.2 mm to bregma) as well as ipsilateral and contralateral M1 tissue samples were prepared as described above, homogenized and immunoblotted.

In order to improve comparability, samples of the same condition were analyzed on the same blotting membrane. Linearity of the signals was ensured by using fluorescence-based immunodetections analyzed by a LI-COR Odyssey System (LI-COR Bioscience, Lincoln, USA).

Primary antibodies used included polyclonal guinea-pig anti-Cobl antibodies (DBY; WB, 1:500) [32], monoclonal mouse anti-actin antibodies (AC-15; Sigma; #A5441; WB, 1:5,000), polyclonal rabbit anti-ß3-tubulin antibodies (Synaptic Systems (Göttingen, Germany); #302

302; WB, 1:2,000), and monoclonal mouse anti-fodrin antibodies (clone AA6; Enzo Life Sciences (Lörrach, Germany); #BML-FG6090; WB, 1:4,000).

Secondary antibodies used were DyLight800-conjugated goat anti-rabbit and anti-mouse antibodies (Thermo Fisher Scientific; #SA5-35571, #SA5-35521) as well as donkey anti-guinea pig antibodies labeled with IRDye680 and IRDye800 (LI-COR Bioscience; #926–68077, #926–32411). For comparison of contra versus ipsi, the contra signal was set to 100%. The ipsi signal was expressed as percent of contra signal (of the same brain sample).

For examinations of Cobl's $Ca^{2+}$- and calpain-dependent proteolysis, mouse brain homogenates were generated in lysis buffer (10 mM HEPES pH 7.4, 1% (v/v) Triton X-100, 0.1 mM $MgCl_2$) containing 10 mM NaCl and 1x Complete protease inhibitor without EDTA (Roche, Basel, Switzerland) and were then either treated with EDTA (1 mM) or incubated with 100 μM $Ca^{2+}$ with and without calpain inhibitor I (CP-1; 15 minutes, 4°C), respectively, and subsequently subjected to anti-Cobl and anti-actin immunoblotting analyses.

## Immunofluorescence staining of mouse brain sections

Free-floating coronal sections (40 μm) from the anterior part of the brain (about +0.8 mm to bregma) at 6 hours reperfusion time were washed with 0.1 M sodium phosphate buffer (pH 7.4) and then permeabilized and blocked by incubation with 5% (v/v) goat serum, 0.25% (v/v) Triton X-100 in 0.1 M sodium phosphate buffer (block solution) for 1 hour at RT. Primary antibody (guinea-pig anti-Cobl antibody DBY; 1:200) labeling was done in block solution for 48 hours at 4°C. The sections were then washed with block solution and subsequently incubated with secondary antibody (AlexaFluor 488-labeled donkey anti-guinea pig (Dianova, Hamburg, Germany)) and with DAPI for 24 hours at 4°C. Finally, the sections were transferred into PBS and mounted with Fluoromount-G (Southern Biotech, Birmingham, USA).

Confocal images were recorded using a Leica TCS SP5 laser confocal microscope. The images used for intensity analysis were recorded with a 10×/0.3 objective. Images were recorded using the *Tiles* mode of the microscope and the automatic stitching of the Zen Software (Zeiss) with an overlap of 10%. Digital images were processed by ImageJ.

Quantitative determinations of anti-Cobl immunolabeling intensities in layer II/III and in layer V in M1 at the contralateral and ipsilateral hemisphere, respectively, of the whole brain images were done by placing 5 rectangular regions of interest (ROIs) at each of the 2 evaluated layers of the 2 M1s in each section. In total, 4 sections from 2 animals were analyzed. The fluorescence signal intensities of the anti-Cobl immunostainings measured in ipsilateral and contralateral ROIs were normalized to the average contralateral intensity of a given section. Quantitative assessments were done with raw image data without any background subtraction.

## Treatments of primary rat cortical neurons and protein isolation from such cultures for quantitative immunoblotting analyses

Rat cortical neurons were prepared as described [57] and cultured until DIV15.

For glutamate stimulations, the primary neuronal cell cultures at DIV15 were washed with Hanks' balanced salt solution (HBSS), and the initial medium was replaced by medium containing 40 μM glutamate. Examined were the excitotoxicity effects induced by 0, 5, 10, 30, and 60 minutes of incubation with glutamate (compare also [40,66]).

In some experiments, inhibitors and antagonists (Calpeptin, 20 μM; CP-1, 20 μM; Chloroquine, 500 μM; Lactacystin, 5 μM; MK801, 50 μM; Ifenprodil, 10 μM; CNQX, 40 μM; AP-5, 100 μM final concentration) were added to the medium (30-minute preincubation; according to [66]). After the preincubations, the cells were then treated with 40 μM glutamate for 30 minutes.

After stimulation, cells were scraped off in lysis buffer containing 10 mM NaCl, 1x Complete protease inhibitor without EDTA (Roche) and 1 mM EDTA. The lysates were sonicated (2 pulses), the homogenates were centrifuged for 10 minutes (4˚C, 1,000 × g), and the protein content was precipitated (20% acetone overnight) at −20˚C. The protein pellet was collected by centrifugation (20,000 × g, 4˚C, 10 minutes), dried, resuspended in 1x SDS sample buffer, and incubated for 10 minutes at 75˚C. The samples were then subjected to western blotting using antibodies described above as well as monoclonal mouse antibodies against PSD95 (6G6-1C9; Abcam (Cambridge, UK); #ab2723; WB, 1:2,000), NR2B (13/NMDAR2B; BD Bioscience (Franklin Lakes, USA); #610416; WB, 1:500), and Arp3 (Abcam; #ab49671; WB, 1:1,000).

The immunosignals were analyzed quantitatively with a LI-COR Odyssey System.

## In vitro reconstitution of Cobl proteolysis by calpain I

HEK293 cells were transfected with plasmids encoding for GFP-Cobl$^{1-713}$ and GFP-Cobl$^{712-1337}$, respectively, which were generated by subcloning from GFP-Cobl full length [30] using an internal Hind III site, and with GFP-Cobl$^{1-408}$, GFP-Cobl$^{406-866}$, GFP-Cobl$^{750-1005}$, and GFP-Cobl$^{1001-1337}$ [30,33,36]. One day after transfection, HEK293 cells were washed with PBS, harvested and lysed by incubation in lysis buffer containing 75 mM NaCl, 1x Complete protease inhibitor without EDTA, and 1 mM EGTA for 30 minutes at 4˚C. Cell lysates were obtained as supernatants from centrifugations at 20,000 × g (20 minutes at 4˚C).

Anti-GFP antibodies (ab290; Abcam) immobilized to protein A-agarose (Santa Cruz Biotechnology (Dallas, USA); #sc-2001) were used to immunocrecipitate the GFP-fusion proteins. After incubation with the HEK293 cell lysates for 3 hours at 4˚C, anti-GFP antibody-associated proteins were isolated by centrifugation at 11,000 × g for 1 minute and washed 2 times with lysis buffer containing 75 mM NaCl. One sample was incubated in cleavage buffer (PBS pH 7.4, 1 mM L-cysteine, 2 mM CaCl$_2$), whereas the others were washed once with cleavage buffer and then incubated at 25˚C with calpain I in 50-μl cleavage buffer for 10 minutes (final concentrations, 0, 0.001, 0.01, and 0.1 U/μl). The cleavage reactions were stopped by adding 4×SDS sample buffer and by denaturing at 95˚C for 5 minutes.

The proteolytic products were immunoblotted with monoclonal mouse anti-GFP antibodies (JL8; Clontech; RRID: AB_10013427) and analyzed by a LI-COR Odyssey System.

## Quantification and statistical analysis

All quantitative data shown represent mean ± SEM and are displayed with overlaid dot plots wherever useful. Tests for normal data distribution and statistical significance analyses were done using Prism 6 software (GraphPad; SCR_002798).

Statistical significances of neuronal morphological analyses in Golgi–Cox-stained M1 tissue samples as well as Cobl-loss-of function and Cobl gain-of-function data in cultures of cortical neurons were tested using Mann–Whitney, 1-way ANOVA with Tukey posttest and 2-way ANOVA with Sidak posttest, and Student $t$ test, respectively. Statistical analyses of Sholl intersections were performed using 2-way ANOVA with Sidak posttest.

Wilcoxon signed rank test was used for changes detected in Cobl protein levels by quantitative westernblotting of lysates of postischemic brains. Quantitative immunofluorescence detections of Cobl protein levels in brain sections were tested using the Mann-Witney test. Moreover, 1-way ANOVA with Dunn posttest was applied to biochemical analyses of Cobl degradation in examinations of glutamate-induced excitotoxicity and the molecular mechanisms involved.

Relative mRNA expression levels in postischemic brains and in sham animals were tested for 1-way ANOVA with Sidak post test.

Infarct volumes were statistically analyzed by unpaired, 2-tailed Student $t$ test.

## Contact for reagent and resource sharing

Further information and requests for resources and reagents should be directed to Michael M. Kessels (Michael.Kessels@med.uni-jena.de).

## Supporting information

**S1 Fig. (Related to Fig 2). The actin nucleator Cobl is degraded by the Ca$^{2+}$-controlled protease calpain during NMDAR-mediated excitotoxicity but *Cobl* KO does not affect the size of the final lesion caused by ischemic stroke induced by MCAO. A, B**, Quantitative immunoblotting analyses of NR2B (**A**) and Arp3 (**B**) in cortical neuronal cultures subjected to different durations of incubation with 40 μM glutamate (Glu). $n = 9$ independent assays and biological samples. **C**, Proteolytic pathways underlying the Cobl decline upon prolonged stimulation with glutamate, as shown by the use of inhibitors against calpain (Calpeptin, CP-1), against lysosomal degradation (Chloroquine) and aginst proteasomal proteolysis (Lactacystin), respectively, in quantitative immunoblotting analyses. $n_{Control} = 10$, $n_{Glu} = 10$, $n_{Glu+Calpeptin} = 10$, $n_{Glu+CP-1} = 10$, $n_{Glu+Chloroquine} = 10$, $n_{Glu+Lactacystin} = 6$ independent biological samples. **D**, Anti-GFP immunoblotting analyses of GFP-Cobl$^{1-408}$, GFP-Cobl$^{406-866}$, GFP-Cobl$^{750-1005}$, and GFP-Cobl$^{1001-1337}$ expressed in HEK293 cells, immunoisolated with anti-GFP antibodies and incubated without and with calpain I (10 minutes, 25°C). Input shows the GFP fusion protein prior to the incubation with calpain concentrations ranging from 0 to 0.1 U/μl. GFP is shown for size comparison. White lines indicate lanes omitted from the blots. Size standards apply to all blots shown. **E**, Lack of effects of inhibitors against open NMDARs (Glu+MK801), against the NR2B subunits of NMDA receptors (Glu+Ifenprodil), against AMPA and kainate receptors (Glu+CNQX), and against NMDARs (Glu+AP-5), respectively, when compared to control and glutamate-induced (30 minutes, 40 μM) excitotoxicity in quantitative anti-β3-tubulin immunoblotting analyses of lysates of neuronal cultures. $n_{control} = 11$, $n_{Glu} = 11$, $n_{Glu+MK-801} = 7$, $n_{Glu+Ifenprodil} = 7$, $n_{Glu+CNQX} = 7$, $n_{Glu+AP-5} = 8$ independent biological samples. Statistical significance calculations, 1-way ANOVA with Dunn posttest (**A–C, E**). $^{***}P < 0.001$. **F,G**, Representative examples of serial sections of brains of WT (**F**) and *Cobl* KO (**G**) mice at 7 days of reperfusion, respectively, as they were used for determinations of MCAO-induced lesion volumes in WT and *Cobl* KO mice in relation to the total brain size. The sections were immunostained with anti-MAP2 antibodies to visualize the lesions. Note the large MAP2-negative lesions (examplarily marked in one section) visible in both WT and *Cobl* KO brains. Bars, 1 mm. **H**, Quantitative determination of lesion volumes caused by 30-minute MCAO in relation to the total brain volume in percent. Note that there was no significant difference in the volume of MCAO-induced lesions when *Cobl* KO brains were compared to WT brains. $n_{WT} = 14$; $n_{KO} = 19$ mice. Statistical significances were calculated using Student $t$ test (n.s.). Data represent means±SEM presented as bar plots overlaid with all individual data points. The numerical data underlying this figure can be found in **S8 Data**. KO, knockout; MCAO, middle cerebral artery occlusion; WT, wild-type.
(TIF)

**S2 Fig. (Related to Fig 3). qPCR analyses demonstrate a transient excess of ipsilateral *Cobl* mRNA expression 6 hours after MCAO. A,B**, Fold change of *Cobl* mRNA at 2 hours and 6 hours reperfusion after MCAO, as determined by qPCR with *Cobl* b (**A**) and *Cobl* c (**B**)

primers, respectively. qPCR data represent the differences between ipsi and contra of *Cobl* (primer sets b and c, respectively) normalized to *Gapdh* (left panel), *Hmbs* (middle panel), and *Tubb3* (right panel), respectively. n $_{MCAO+2h}$ = 8; n $_{MCAO+6h}$ = 8 brain samples and mice. **C,D**, Fold change of *CaM 1* (**C**) and *Calpain I* (**D**) mRNA levels at 2 hours, 6 hours, 12 hours, and 48 hours reperfusion time after 30-minute MCAO. Data represent ratios of the differences between ipsi and contra of *CaM 1* and *Calpain I* levels, respectively, normalized to *Gapdh*, *Hmbs*, and *Tubb3*, respectively. **E**, Fold changes of *Cobl* mRNA levels in M1 tissue samples (ipsi versus contra) 6 hours and 12 hours after MCAO. The data are again normalized against 3 different genes (*Gapdh*, *Hmbs*, and *Tubb3*). **A–D**, n $_{MCAO+2h}$ = 8; n $_{MCAO+6h}$ = 8; n $_{MCAO+12h}$ = 9; n $_{MCAO+48h}$ = 6 brain samples and mice. **E**, n$_{MCAO+6h}$ = 7; n$_{MCAO+12h}$ = 8 M1 samples and mice. Data, mean ± SEM. Statistical significances (ipsi versus contra) were calculated using 1-way ANOVA with Sidak posttest (**A–E**), respectively. $^{**}P < 0.01$; $^{***}P < 0.001$. The numerical data underlying this figure can be found in **S9 Data**. MCAO, middle cerebral artery occlusion; qPCR, quantitative PCR.
(TIF)

**S3 Fig. (Related to Fig 4). No change in dendritic arbor complexity subsequent to ischemic stroke in layer II/III and layer V neurons in the contralateral motor cortex. A,B,** Representative micrographs of adjacent serial coronal sections of brains at ~bregma +0.8 mm from WT mice (age, 3 to 4 months) that were subjected to 30-minute MCAO. **A**, Anti-MAP2 immunostaining. **B**, Golgi–Cox staining. The lesion caused by the infarct can be determined based on the lack of anti-MAP2 detection (outlined in red). Blue framing indicates the adjacent region M1 used for morphological analysis in corresponding Golgi–Cox-stained sections. **C-E**, Representative images of Golgi–Cox-stained (left panels) and Imaris-reconstructed (right panels) layer II/III pyramidal neurons of M1 from the contralateral side of sham-treated mice (**C**) and of mice subjected to 30-minute MCAO analyzed after 6 hours (**D**) and 24 hours (**E**) reperfusion times. The position of the cell bodies are marked by a gray dot. Scale bars, 30 μm. **F–M**, Quantitative determinations of dendritic branching points (**F, J**), dendritic terminal points (**G, K**), total dendritic tree length (**H, L**), and Sholl intersections (**I, M**) of dendritic trees in layer II/III (**F–I**) and layer V of M1 (**J–M**). Note that all parameters of dendritic complexity at the contralateral side remained unchanged in comparison to sham-treated mice at both 6 hours and 24 hours after MCAO. Layer II/III: n$_{Sham\ contra}$ = 57; n$_{MCAO+6h\ contra}$ = 18; n$_{MCAO+24h\ contra}$ = 17; n$_{Sham\ ipsi}$ = 60 neurons. Layer V: n$_{Sham\ contra}$ = 60; n$_{MCAO+6h\ contra}$ = 22; n$_{MCAO+24h\ contra}$ = 25; n$_{Sham\ ipsi}$ = 47 neurons from 3 mice each for the 2 MCAO conditions (for ipsi data from the same mice, see **Fig 4**) and from 6 mice for sham controls (contralateral and ipsilateral; sham ipsi data as in **Fig 4** for comparison). Quantitative data represent mean ± SEM. Statistical significance calculations, 1-way ANOVA with Tukey posttest (**F–H, J–L**) and 2-way ANOVA with Sidak posttest for Sholl analysis (**I, M**), respectively (all n.s.). The numerical data underlying this figure can be found in **S10 Data**. MCAO, middle cerebral artery occlusion; WT, wild-type.
(TIF)

**S4 Fig. (Related to Fig 5). MCAO-induced defects in dendritic arborization manifest in both apical and basal dendrites of layer II/III pyramidal neurons of M1. A–C**, Individual Imaris-reconstructed cells from **Fig 5A–5C** (right panels) showing ipsilateral layer II/III neurons from M1 of WT mice subjected to sham treatment (**A**) and 30-minute MCAO with 24 hours and 4 days reperfusion time (**B, C**), respectively, for differential analyses of basal (**D–H**) and apical dendritic parameters (**I–K**). The apical and basal dendritic parts are depicted in **A–C**. The positions of the cell bodies are marked by a gray dot. Scale bars, 30 μm. **D–K**, Quantitative determinations of dendritic branching points, terminal points, total dendritic tree length, and Sholl intersections of basal dendrites (**D–H**) and related analyses for apical dendrites (**I–**

**K**). Note that MCAO-induced defects occur in both apical and basal dendrites 24 after MCAO and also show full recovery in both apical and basal dendritic arbors after 4 days reperfusion. $n_{Sham+24h} = 18$; $n_{MCAO+24h} = 43$; $n_{Sham+4d} = 35$; $n_{MCAO+4d} = 32$ neurons from 6 mice for the MCAO+24h group and 3 mice for the MCAO+4d group and from 3 mice for each sham control group (ipsi). Quantitative data represent mean ± SEM. Statistical significance calculations, Mann–Whitney (**D–F, I–K**) and 2-way ANOVA with Sidak posttest for Sholl analysis (**G, H**), respectively. $^*P < 0.05$; $^{**}P < 0.01$; $^{***}P < 0.001$. The numerical data underlying this figure can be found in **S11 Data**. MCAO, middle cerebral artery occlusion; WT, wild-type.
(TIF)

**S5 Fig. (Related to Fig 5). The dendritic arborization of neurons at the contralateral side is not affected by the MCAO-induced dendritic regrowth processes occurring simultaneously at the ipsilateral side. A–L**, Quantitative determinations of dendritic arborization parameters of layer II/III neurons (**A–F**) and layer V neurons (**G–L**) in the contralateral M1. Note that, at the contralateral side, dendritic branching points (**A, G**), dendritic terminal points (**B, H**), total dendritic tree length (**C, I**) and Sholl intersections (**D–F, J–L**) at different reperfusion times all remained similar to their respective sham controls and at the same level for all 3 time points (24 hours, 4 days, and 7 days). Layer II/III: $n_{Sham+24h} = 29$; $n_{MCAO+24h} = 53$; $n_{Sham+4d} = 38$; $n_{MCAO+4d} = 29$; $n_{Sham+7d} = 46$; $n_{MCAO+7d} = 49$ neurons from 3 mice for the sham+24h, sham+4d and MCAO+4d groups, and from 6 mice for the sham+7d, MCAO+24h, and MCAO+7d groups (3 to 4 months of age). Layer V: $n_{Sham+24h} = 17$; $n_{MCAO+24h} = 57$; $n_{Sham+4d} = 30$; $n_{MCAO+4d} = 30$; $n_{Sham+7d} = 57$; $n_{MCAO+7d} = 50$ neurons from 3 mice of each group. For corresponding ipsilateral data, see **Fig 5**. Data, mean ± SEM. Statistical significances were calculated using 2-way ANOVA with Sidak posttest (all n.s.). The numerical data underlying this figure can be found in **S12 Data**. MCAO, middle cerebral artery occlusion.
(TIF)

**S6 Fig. (Related to Fig 7). *Cobl* KO mice show no MCAO-induced dendritic alterations of the dendritic arbor at the contralateral side. A–L**, Quantitative determinations of dendritic arborization parameters of layer II/III neurons (**A–F**) and layer V neurons (**G–L**) at the contralateral side in M1 of *Cobl* KO mice. Note that dendritic branching points (**A, G**), dendritic terminal points (**B, H**), total dendritic tree length (**C, I**), and Sholl intersections (**D–F, J–L**) at 24 hours, 4 days, and 7 days reperfusion times after 30-minute MCAO at the juxtaposed side all remained unchanged and similar to their corresponding sham controls. Layer II/III: $n_{Sham+24h} = 28$; $n_{MCAO+24h} = 32$; $n_{Sham+4d} = 33$; $n_{MCAO+4d} = 28$; $n_{Sham+7d} = 31$; $n_{MCAO+7d} = 36$ neurons from 3 mice of each group. Layer V: $n_{Sham+24h} = 34$; $n_{MCAO+24h} = 33$; $n_{Sham+4d} = 28$; $n_{MCAO+4d} = 34$; $n_{Sham+7d} = 26$; $n_{MCAO+7d} = 27$ neurons from 3 mice of each group. For corresponding ipsilateral data, see **Fig 7**. Data represent mean ± SEM. Statistical significances were calculated using 2-way ANOVA with Sidak posttest (all n.s.). The numerical data underlying this figure can be found in **S13 Data**. KO, knockout; MCAO, middle cerebral artery occlusion.
(TIF)

**S7 Fig. (Related to Fig 7). The defects in ipsilateral dendritic arbor repair observed in Cobl KO mice occur in both apical and basal dendrites of layer II/III pyramidal neurons of M1. A–C**, Individual Imaris-reconstructed cells from **Fig 7A–7C** (right panels) showing ipsilateral layer II/III neurons from M1 of *Cobl* KO mice subjected to sham treatment (**A**) and 30-minute MCAO with 24 hours and 4 days reperfusion time (**B, C**), respectively, for differential analyses of basal (**D–H**) and apical dendritic parameters (**I–K**). The apical and basal dendritic parts are depicted in **A–C**. The somas are marked by a gray dot. Scale bars, 30 μm. **D–K**, Quantitative determinations of dendritic branching points, terminal points, total dendritic tree length, and

Sholl intersections of basal dendrites (**D–H**) and related analyses for apical dendrites (**I–K**). Note that MCAO-induced defects occurring in both apical and basal dendrites 24 after MCAO, in contrast to WT (see **Fig 5**), do not show any recovery after 4 days reperfusion upon *Cobl* KO. $n_{Sham+24h} = 21$; $n_{MCAO+24h} = 35$; $n_{Sham+4d} = 30$; $n_{MCAO+4d} = 29$ neurons neurons from 3 mice for each MCAO and sham control group (ipsi). Quantitative data represent mean ± SEM. Statistical significance calculations, Mann–Whitney (**D–F, I–K**) and 2-way ANOVA with Sidak posttest for Sholl analysis (**G, H**), respectively. $^*P < 0.05$; $^{**}P < 0.01$; $^{***}P < 0.001$. The numerical data underlying this figure can be found in **S14 Data**. KO, knockout; MCAO, middle cerebral artery occlusion; WT, wild-type.
(TIF)

**S1 Data. Numerical data underlying the quantitative data panels of Fig 1.** This Excel file contains all numerical information of all data panels in **Fig 1** organized in form of subfolders. The data include mean, SEM, *n* number, and all individual data points.
(XLSX)

**S2 Data. Numerical data underlying the quantitative data panels of Fig 2.** This Excel file contains all numerical information of all data panels in **Fig 2** organized in form of subfolders. The data include mean, SEM, *n* number, and all individual data points.
(XLSX)

**S3 Data. Numerical data underlying the quantitative data panels of Fig 3.** This Excel file contains all numerical information of all data panels in **Fig 3** organized in form of subfolders. The data include mean, SEM, *n* number, and all individual data points.
(XLSX)

**S4 Data. Numerical data underlying the quantitative data panels of Fig 4.** This Excel file contains all numerical information of all data panels in **Fig 4** organized in form of subfolders. The data include mean, SEM, *n* number, and all individual data points.
(XLSX)

**S5 Data. Numerical data underlying the quantitative data panels of Fig 5.** This Excel file contains all numerical information of all data panels in **Fig 5** organized in form of subfolders. The data include mean, SEM, *n* number, and all individual data points.
(XLSX)

**S6 Data. Numerical data underlying the quantitative data panels of Fig 6.** This Excel file contains all numerical information of all data panels in **Fig 6** organized in form of subfolders. The data include mean, SEM, *n* number, and all individual data points.
(XLSX)

**S7 Data. Numerical data underlying the quantitative data panels of Fig 7.** This Excel file contains all numerical information of all data panels in **Fig 7** organized in form of subfolders. The data include mean, SEM, *n* number, and all individual data points.
(XLSX)

**S8 Data. Numerical data underlying the quantitative data panels of S1 Fig.** This Excel file contains all numerical information of all data panels in **S1 Fig** organized in form of subfolders. The data include mean, SEM, *n* number, and all individual data points.
(XLSX)

**S9 Data. Numerical data underlying the quantitative data panels of S2 Fig.** This Excel file contains all numerical information of all data panels in **S2 Fig** organized in form of subfolders.

The data include mean, SEM, *n* number, and all individual data points.
(XLSX)

**S10 Data. Numerical data underlying the quantitative data panels of S3 Fig.** This Excel file contains all numerical information of all data panels in S3 Fig organized in form of subfolders. The data include mean, SEM, *n* number, and all individual data points.
(XLSX)

**S11 Data. Numerical data underlying the quantitative data panels of S4 Fig.** This Excel file contains all numerical information of all data panels in S4 Fig organized in form of subfolders. The data include mean, SEM, *n* number, and all individual data points.
(XLSX)

**S12 Data. Numerical data underlying the quantitative data panels of S5 Fig.** This Excel file contains all numerical information of all data panels in S5 Fig organized in form of subfolders. The data include mean, SEM, *n* number, and all individual data points.
(XLSX)

**S13 Data. Numerical data underlying the quantitative data panels of S6 Fig.** This Excel file contains all numerical information of all data panels in S6 Fig organized in form of subfolders. The data include mean, SEM, *n* number, and all individual data points.
(XLSX)

**S14 Data. Numerical data underlying the quantitative data panels of S7 Fig.** This Excel file contains all numerical information of all data panels in S7 Fig organized in form of subfolders. The data include mean, SEM, *n* number, and all individual data points.
(XLSX)

**S15 Data. Compilation of uncrobbed western blots shown in Figs 1 and 2 and S1.** This 3-page PDF contains all western blots shown in **Figs 1 and 2** and **S1** as an uncropped images. Lanes shown in the respective figure panel (see panel labeling) are framed in green, and lanes not shown are crossed out. Information on the samples (conditions) and on the immunodetections is labeled as in **Figs 1 and 2** and **S1**. Note that some blots where physically cut for the multiple detects and thus do not include the full range of protein sizes (see standard lanes).
(PDF)

## Acknowledgments

We thank I. Ingrisch and M. Öhler for excellent technical support and L. Schwintzer, J. Tröger, M. Izadi as well as N. Haag for their generous support with plasmids, practical help, and mice, respectively.

## Author Contributions

**Conceptualization:** Michael M. Kessels, Christiane Frahm, Britta Qualmann.

**Data curation:** Yuanyuan Ji, Dennis Koch, Christiane Frahm.

**Formal analysis:** Yuanyuan Ji, Dennis Koch, Christiane Frahm.

**Funding acquisition:** Michael M. Kessels, Christiane Frahm, Britta Qualmann.

**Investigation:** Yuanyuan Ji, Dennis Koch.

**Methodology:** Yuanyuan Ji, Madlen Günther.

**Project administration:** Michael M. Kessels, Britta Qualmann.

**Resources:** Jule González Delgado, Otto W. Witte, Michael M. Kessels, Britta Qualmann.

**Supervision:** Michael M. Kessels, Christiane Frahm, Britta Qualmann.

**Validation:** Michael M. Kessels, Britta Qualmann.

**Visualization:** Yuanyuan Ji, Michael M. Kessels.

**Writing – original draft:** Yuanyuan Ji, Michael M. Kessels, Britta Qualmann.

**Writing – review & editing:** Yuanyuan Ji, Dennis Koch, Jule González Delgado, Michael M. Kessels, Christiane Frahm, Britta Qualmann.

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
