## [Editor Report · Decision Letter 0]

18 Aug 2021

Dear Dr Kessels, 

Thank you for submitting your revised manuscript entitled "Post-stroke dendritic arbor regrowth – a cortical repair process requiring the actin nucleator Cobl" for consideration as a Research Article by PLOS Biology.

Your revision has now been evaluated by the PLOS Biology editorial staff as well as by an academic editor with relevant expertise and I am writing to let you know that we would like to send your submission out for external peer review.

Please re-submit your manuscript within two working days, i.e. by Aug 20 2021 11:59PM.

Kind regards,

Lucas Smith

Associate Editor

PLOS Biology

lsmith@plos.org

---

## [Decision Letter · Decision Letter 1]

8 Oct 2021

Dear Dr Kessels,

Thank you very much for submitting a revised version of your manuscript "Post-stroke dendritic arbor regrowth - a cortical repair process requiring the actin nucleator Cobl" for consideration as a Research Article at PLOS Biology. This revised version of your manuscript has been evaluated by the PLOS Biology editors, the Academic Editor and the original reviewers.

As you will see, the reviewers appreciate the improvements you have made to the manuscript and Reviewers 1 and 2 as largely satisfied. We note that Reviewer 1 suggested accept with no additional comments however Reviewer 2 highlights a number of minor points which will need to be addressed. Moreover, Reviewer 3 raises an important lingering concern and thinks that it would be important to demonstrate a reduction of Cobl in the neurons of layers 11/111 of M1. While we appreciate that you highlighted issues with antibodies, we think that this point should be experimentally addressed with staining, qRT-PCR, western blot, FISH or other techniques.

Reviewer 4 has commented that you should use a neuron specific Cobl KO model, and that without this s/he questions whether the study is at the level of PLOS Biology. While we appreciate that the additional studies suggested by Reviewer 4 would be interesting and would strengthen the study, after a careful discussion within the team and with the Academic Editor, we do not think that these analyses would be required for publication in PLOS Biology at this stage.

In light of the reviews, we will not be able to accept the current version of the manuscript, but we would welcome re-submission of a much-revised version that takes into account the reviewers' comments. We cannot make any decision about publication until we have seen the revised manuscript and your response to the reviewers' comments. Your revised manuscript may be sent for further evaluation by the reviewers.

We expect to receive your revised manuscript within 3 months. 

**IMPORTANT - SUBMITTING YOUR REVISION**

*Re-submission Checklist*

*Published Peer Review*

*PLOS Data Policy*

*Blot and Gel Data Policy*

Sincerely,

Lucas Smith

Associate Editor

PLOS Biology

lsmith@plos.org

REVIEWS:

Reviewer #1:

Reviewer #2: I appreciated the new data in Fig 1 showing changes in Cobl mRNA and protein expression specifically in the intact ipsilateral M1 (rather than whole hemisphere) which, is an important part of the story with respect to dendritic retraction and growth of pyramidal neurons. The authors have also done a much more thorough job placing the present findings in the context of past literature. In addition, the authors now provide more direct evidence between dendritic arbor remodelling and Cobl regulation with gain and loss of function experiments. Thus, the manuscript's primary conclusion has much stronger supporting data and is convincing. I fully support publishing this paper, although I have a few minor comments for the authors to consider.

- The lack of observed changes in dendritic arbors in the contralateral hemisphere after MCAO is important (data in Supp Fig. 3 and 5) and is consistent with previous work from Johnston, Denizet et al., Cerebral Cortex, 2013.

- I was surprised the authors didn't combine Supp Fig 6 with main fig 6, since the data show growth promoting effect of Cobl overexpression (Supp Fig 6) while Cobl RNAi decreases dendritic branching. These 2 pieces of data when combined are quite strong, perhaps they could combine these into 1 main figure?

- Line 300-301: sentence is confusing "may not do so not only…"

Reviewer #3: It is important to show the reduction of Cobl in the neurons of layers 11/111 of M1 where the dendritic arborization is reduced

Reviewer #4: The manuscript "Post-stroke dendritic arbor regrowth - a cortical repair process requiring the actin nucleator Cobl" improved in the revision process. I am sorry that I probably did not make it sufficiently clear when I wrote in my first comments "that it appears that the paper is, unfortunately, not on the level of PLoS Biology" and that "I have some specific comments below that may help the authors to prepare a submission to another journal". 

In principle, I would expect for a PLoS Biology paper something on a different scale nowadays. For the claims the authors make they would need a KO model where Cobl is genetically ablated only in neurons and do the stroke work in such a model. This was clearly out of the scope of their work. And this is why I initially rejected the paper and gave some suggestions to improve the quality of the paper to submit it to another journal. Now the paper did improve. But it still is not on the level of a PLoS paper. It just simply would in my opinion not be a PLoS Biology without such more rigorous analysis of Cobl function in vivo, as mentioned above.

---

## [Editor Report · Decision Letter 2]

5 Nov 2021

Dear Dr Kessels,

Thank you for submitting your revised Research Article entitled "Post-stroke dendritic arbor regrowth - a cortical repair process requiring the actin nucleator Cobl" for publication in PLOS Biology. Your revised manuscript has now been evaluated by the editorial team and by the Academic Editor, and we think you have adequately addressed the points we highlighted in the last round of review. 

While we are satisfied with the revision, before we can editorially accept your manuscript and proceed to our production checks, we have two last editorial requests which need to be addressed. **Please address the following requests in a revised manuscript:

1) DATA REQUEST: Thank you for providing, as a supplementary file, the data underlying your figures ("Figure1-7_S1-S7_All_Numerical_Data.xlsx"). Looking through this file, I see that the data is represented as the mean + SEM. Unfortunately, this does not satisfy our data availability requirements (http://journals.plos.org/plosbiology/s/data-availability)

Please update this file, to include all of the individual quantitative observations that underlie the data summarized in the figures and results of your paper.

Please also ensure that figure legends in your manuscript include information on where the underlying data can be found, and ensure that this supplemental data file/s has a legend. For example, in each figure legend (including supplementary figures), you can add the sentence "the data underlying this figure can be found in S1_data"

2) SCALE BARS: Please provide a scale bar for Figure 1Q

3) TITLE: After a bit of discussion within the team, we think that the current title may benefit from being simplified a bit. If you agree, we would suggest something along the lines of "Post-stroke dendritic arbor regrowth requires the actin nucleator Cobl".

We expect to receive your revised manuscript within two weeks. 

*Published Peer Review History*

*Early Version*

Sincerely,

Lucas Smith, Ph.D.,

Associate Editor,

lsmith@plos.org,

PLOS Biology

---

## [Editor Report · Decision Letter 3]

16 Nov 2021

Dear Dr Kessels,

Thank you very much for addressing our last editorial requests with your last revision. On behalf of my colleagues and the Academic Editor, Richard Daneman, I am pleased to say that we can in principle accept your Research Article "Post-stroke dendritic arbor regrowth requires the actin nucleator Cobl" for publication in PLOS Biology.

Please note, that before publication you will need to address any remaining formatting and reporting issues which will be detailed in an email that will follow this letter (you will usually receive this from our production office within 2-3 business days, during which time no action is required from you). Please note that we will not be able to schedule your manuscript for publication until you made have any requested changes.

PRESS

Sincerely, 

Lucas Smith, Ph.D. 

Senior Editor 

PLOS Biology

lsmith@plos.org